

# Spatio-temporal patterns of the effects of precipitation variability
# and land use/cover changes on long-term changes in sediment yield
# in the Loess Plateau, China
Guangyao Gao[1,2], Bojie Fu[1,2], and Murugesu Sivapalan[3,4]
[1]State Key Laboratory of Urban and Regional Ecology, Research Center for
Eco-Environmental Sciences, Chinese Academy of Sciences, Beijing 100085, China
[2]Joint Center for Global Change Studies, Beijing 100875, China
[3]Department of Geography and Geographic Information Science, University of Illinois at
Urbana-Champaign, Champaign, Illinois, USA
[4]Department of Civil and Environmental Engineering, University of Illinois at
Urbana-Champaign, Urbana, Illinois, USA
*Correspondence to:* Guangyao Gao (gygao@rcees.ac.cn)
**Abstract**
Within China's Loess Plateau there have been concerted revegetation efforts and
engineering measures over the last 50 years aimed at reducing soil erosion and land
degradation. As a result, annual streamflow, sediment yield and sediment concentration
have all decreased considerably. Human induced land use/cover change (LUCC) was the
dominant factor, contributing over 70% of the sediment load reduction, with reductions of
annual precipitation contributing the remaining 30%. In this study, we use data on 50-year
time series (1961-2011), showing decreasing trends in the annual sediment loads of fifteen





catchments, to generate spatio-temporal patterns in the effects of LUCC and precipitation
variability on sediment yield. The space-time variability of sediment yield was expressed as
a product of two factors representing: (i) effect of precipitation (spatially variable) and (ii)
fraction of treated land surface area (temporally variable). Under minimal LUCC, annual
sediment yield varied linearly with precipitation, with the precipitation-sediment load
relationship showing coherent spatial patterns amongst the catchments. On the other hand,
the effect of LUCC is expressed in terms of a sediment coefficient, i.e., ratio of annual
sediment yield to annual precipitation, which is equivalent to the slope of the sediment
yield-precipitation relationship. Sediment coefficients showed a steady decrease over the
study period, following a linear decreasing function of the fraction of treated land surface
area. In this way, the study has brought out the separate roles of precipitation variability
and LUCC in controlling spatio-temporal patterns of sediment yield at catchment scale.
**Keywords:** Loess Plateau, sediment yield, land use/land cover change, climate change,
precipitation variability
**1   Introduction**
Streamflow and sediment transport are important controls on biogeochemical processes
that govern ecosystem health in river basins (Syvitski, 2003). Changes in soil erosion on
landscapes and the resulting changes in sediment transport rates in rivers have great
environmental and societal consequences, particularly since they can be brought about by
climatic changes and human induced land use/cover changes (LUCC) (Syvitski, 2003;





Beechie et al., 2010). Understanding the dominant mechanisms behind such changes at
different time and space scales is crucial to the development of strategies for sustainable
land and water management in river basins (Wang et al., 2016).
In recent decades, streamflows and sediment yields in large rivers throughout the world
have undergone substantial changes (Milly et al., 2005; Nilsson et al., 2005; Milliman et al.,
2008; Cohen et al., 2014). Notable decreases in sediment yields have been observed in
approximately 50% of the world's rivers (Walling and Fang, 2003; Syvitski et al., 2005).
Many studies have investigated the dynamics of streamflows and sediment yields at
different spatial and temporal scales (Mutema et al., 2015; Song et al., 2016; Gao et al.,
2016; Tian et al., 2016). In addition to climate variability, LUCC, soil and water
conservation measures (SWCM) and construction of reservoirs and dams have substantially
contributed to the sediment load reductions (Walling, 2006; Milliman et al., 2008; Wang et
al., 2011). While previous studies have certainly provided valuable insights into the
streamflow and sediment load changes, the distinctive roles of LUCC and precipitation
variability in changing sediment loads still need further investigation in large domains and
across gradients of climate and land surface conditions (Walling, 2006; Mutema et al.,
2015). A particularly useful approach to the development of generalizable understanding of
the effects of precipitation variability and LUCC is a comparative analysis approach
focused on extracting spatio-temporal patterns of sediment yields based on observations in
multiple locations within the same region, or even across different regions. This is
especially valuable and crucial in areas with severe soil erosion and fragile ecosystems, e.g.,
the Loess Plateau (LP) in China. This is the motivation for the work presented in this paper.





The LP lies in the middle reaches of the Yellow River (YR) Basin, and contributes
nearly 90% of the YR sediment (Wang et al., 2016). The historically severe soil erosion in
the LP is due to sparse vegetation, intensive rainstorms, erodible loessial soil, steep
topography and a long agricultural history (Rustomji et al., 2008). To control such severe
soil erosion, several SWCM including terrace and check-dam construction, afforestation
and pasture reestablishment have been implemented since the 1950s (Yao et al., 2011; Zhao
et al., 2016). A large ecological restoration campaign, the Grain-for-Green (GFG) project
by converting farmland on slopes exceed 15° to forest and pasture lands, was implemented
in 1999 (Chen et al., 2015). Furthermore, the climate in the LP region has been showing
both warming and drying trends (i.e., increased potential evapotranspiration and reduced
precipitation) since the 1950s (Zhang et al., 2016).
These substantial LUCC have notably altered the hydrological regimes in the LP
combined with the climate change. Consequently, the sediment yields within the LP have
showed a predictable decline trend over the past 60 years (Zhao et al., 2016), resulting in
approximately a 90% decrease of sediment yield in the YR (Miao et al., 2010, 2011; Wang
et al., 2016). Many other studies have detected the influences of LUCC and precipitation
variability on sediment load changes within the LP. Rustomji et al. (2008) estimated that
the contributions of catchment management practices to the decrease of annual sediment
yield ranged between 64% and 89% for eleven catchments in the LP during 1950s-2000.
Zhao et al. (2016) examined the spatio-temporal variation of sediment yield from 1957 to
2012 across the LP. Zhang et al. (2016) pointed out that the combined effects of climate
aridity, engineering projects and vegetation cover change have induced significant



reductions of sediment yield between 1950 and 2008. Wang et al. (2016) found that
engineering measures for soil and water conservation were the main factors for the
sediment load decrease between 1970s-1990s, but large-scale vegetation restoration
campaigns also played important role in reducing soil erosion since the 1990s.

In terms of the results of these previous studies, it is now generally accepted that the

largest reductions of sediment yield within the LP were resulted from LUCC. However, this
is general knowledge covering the whole region, and given the significant variability of
climate and catchment characteristics across the LP (Sun Q et al., 2015; Sun W et al., 2015),
it is important to go further and explore how these might affect spatio-temporal patterns of
sediment yield. Exploration of these patterns is important for sustainable ecosystem
restoration and water resources planning and management within the LP. They also will
serve as the basis for future research aimed at the development of more generalizable
understanding of landscape and climate controls on sediment yields at the catchment scale.
The specific objectives of this study therefore are to: (1) attribute the temporal changes
in sediment yield to changes in both precipitation variability and LUCC over the entire
study period (1961-2011) within the middle part of the LP, (2) extract spatio-temporal
trends in sediment yields on the basis of annual sediment yield data from 15 catchments
within the region, (3) separate the contributions of precipitation variability and fractional
area of LUCC to the observed spatio-temporal patterns of sediment yields, and pave the
way for more detailed process-based studies in the future.
**2   Materials and methods**
**2.1 Study area**



This study is conducted in the central region of the LP, from the Toudaoguai to Longmen
hydrological stations in the mainstream of the YR (Fig. 1). This area is usually referred to
as the Coarse Sandy Hilly Catchments (CSHC) region. The main stream that flows through
the CSHC region is 733 km long and covers an area of $12.97 \times 10^4$ km$^2$. The CSHC region
accounts for 14.8% of the entire YR Basin, but supplies over 70% of total sediment load in
the YR, especially coarse sand (Rustomji et al., 2008). The CSHC region is characterized
by arid to semi-arid climate conditions. The annual precipitation in the CSHC region during
1961-2011 is 437 mm on average, and varied from 580 mm in the southeast to lower than
300 mm in the northwest (McVicar et al., 2007). The precipitation that occurs during the
flood season (June-September) is usually in the form of rainstorms with high intensity and
accounts for 72% of the annual rainfall total. Correspondingly, about 45% of the annual
runoff and 88% of the annual sediment yield within the CSHC region are produced during
the flood season. The northwestern part of the CSHC is relatively flat while the
southeastern part is more finely dissected (Rustomji et al., 2008).
Fourteen main catchments along the north-south transect within the CSHC study area
were chosen for study (Fig. 1). These catchments account for 57.4% of the CSHC area, and
contribute about 70% and 72% of streamflow and sediment load of the overall CSHC,
respectively. Characteristics of these catchments are shown in Table 1. It can be seen that
the catchments present strong climate and land surface gradients. The catchments in the
northwestern part (#1-6) have relatively lower mean annual precipitation (380 mm$< \overline{P} <$445
mm, where $\overline{P}$ is mean annual precipitation over 1961-2011) and low vegetation cover
(0.32<LAI<0.37, where LAI is the leaf area index), while the corresponding values for





catchments in the southeastern part (#7-14) are 470-570 mm and 0.63<LAI<2.16,
respectively. The entire CSHC region is considered as an additional "catchment" and it is
also examined. The streamflow and sediment load for the whole CSHC region was equal to
the differences of value between the Toudaoguai and Longmen gauging station. Fig. 2
shows the changes of annual precipitation, streamflow and sediment load for the whole
CSHC region during 1961-2011.
**2.2 Data**
Monthly streamflow and sediment load data during 1961-2011 were provided by the
Yellow River Conservancy Commission of China. Daily rainfall data from 1961 to 2011 at
66 meteorological stations in and around the CSHC region were obtained from the National
Meteorological Information Center of China. The spatially average of rainfall data were
determined by the co-kriging interpolation algorithm with input of the DEM. With the
hydro-meteorological data, annual precipitation, $P$ [mm], streamflow, $Q$ [mm], specific
sediment yield defined as $SSY=S/A$ [t km$^{-2}$], where $S$ is sediment load, t, $A$ is the drainage
area of the hydrological station, km$^2$, sediment concentration defined as $SC=S/(Q.A)$ [kg
m$^{-3}$] and the sediment coefficient defined as $C_s=SSY/P$ [t km$^{-2}$ mm$^{-1}$] for each catchment
were estimated.

The land use information as at 1986, 1997 and 2010 was determined with Landsat TM

remote sensing images at a spatial resolution of 30 m. Six land use types were classified,
i.e., forestland, cropland, grassland, construction land, water body, and wasteland. The
annual LAI data during 1982-2011 were obtained from the Global Inventory Modelling and
Mapping Studies-Advanced Very High Resolution Radiometer (GIMMS AVHRR) data set





(http://www.glcf.umd.edu/data/lai/) which has a spatial resolution of 8 km (resampled to 1
km) and temporal frequencies of 15 day. Vegetation cover in the summer or autumn of
1978, 1998 and 2010 was determined with Landsat MSS, Landsat TM and HJ CCD which
has a spatial resolution of 56 m, 30 m and 30 m, respectively. The total areas impacted by
the various SWCM (i.e., afforestation, grass plantation, terraces and check-dams)
during1960s-2000s were obtained from Yao et al. (2011).
**2.3 Methods**
**2.3.1 Trend test**
The non-parametric Mann-Kendall (M-K) test method proposed by Mann (1945) and
Kendall (1975) was used to determine the significance of the trends in annual
meteorological and hydrological time series. A precondition for using the MK test is to
remove the serial correlation of climatic and hydrological series. In this study, the
trend-tree pre-whitening (TFPW) method of Yue and Wang (2002) was used to remove the
auto-correlations before the trend test. A $Z$ statistic was obtained from the M-K test on the
whitened series. A negative value of $Z$ indicates a decrease trend, and vice versa. The
magnitude slope of the trend ($\beta$) was estimated by (Sen, 1968; Hirsch et al., 1982):
$$\beta = \text{Median}\left[\frac{x_j - x_i}{j - i}\right] \quad \text{for all } i<j \tag{1}$$

where $x_i$ and $x_j$ are the sequential data values in periods $i$ and $j$, respectively.
**2.3.2 Attribution analysis of changes in sediment load**
The time-trend analysis method was used to determine the quantitative contributions of
LUCC and precipitation variability to sediment load changes. This method is primarily
designed to determine the differences in hydrological time series between different periods



(reference and validation periods) with different LUCC conditions (Zhang et al., 2011). In
this method, the regression equation between precipitation and sediment load is developed
and evaluated during the reference period, and the established equation is then used to
estimate sediment load during the validation period. The difference between measured and
predicted sediment loads during the validation period represents the effects of LUCC, and
the residual changes are caused by precipitation variability. The governing equations of the
time-trend analysis method can be expressed as:
$$S_1 = f(P_1) \tag{2}$$
$$S_2{}' = f(P_2) \tag{3}$$
$$\Delta S^{\mathrm{LUCC}} = \overline{S_2} - \overline{S_2'} \tag{4}$$
$$\Delta S^{\mathrm{Pre}} = \left(\overline{S_2} - \overline{S_1}\right) - \Delta S^{\mathrm{LUCC}} \tag{5}$$
where $S'$ is the predicted sediment load, subscripts 1 and 2 indicate the reference and
validation periods, respectively. $\overline{S_1}$ and $\overline{S_2}$ represent mean measured sediment load during
the reference and validation periods, respectively, and $\overline{S_2'}$ represents mean predicted
sediment load during the validation period. $\Delta S^{\mathrm{LUCC}}$ and $\Delta S^{\mathrm{Pre}}$ are sediment load changes
during the validation period associated with LUCC and precipitation variability,
respectively.

In this study, the full data period of 1961-2011 was divided into three phases

(1961-1969, 1970-1999 and 2000-2011). The first period was considered the reference
period as the effects of human activities were slight and could be mostly ignored (Wang et
al., 2016). A linear function was used to develop precipitation-sediment load relationship
during the reference period. During the second stage, numerous SWCM were implemented.



For the third stage, a large ecological restoration campaign (GFG project) was launched in

1999.

**3   Results and discussion**
**3.1  Changes of land use/cover**
The CSHC region has undergone extensive LUCC caused by the implementation of
SWCM and vegetation restoration projects (e.g., the GFG project). Fig. 3a shows the
distribution of land use types of the CSHC region in 1986, 1997 and 2010. More than 90%
of the whole area is occupied by the cropland, forestland and grassland. The area of
cropland decreased by 19.47% and forestland increased by 59.65%, and there was no
obvious change for the area of grassland from 1986-2010. The majority of changes
occurred during 1997-2010 due to the GFG (reforestation) project (18.58% decrease and
40.71% increase for cropland and forestland, respectively). From 1986 to 2010, the water
body area increased by 88.05% due to construction of reservoirs, and construction land
increased by about twenty times because of the urbanization and extensive infrastructure
construction.

The SWCM implemented in the LP included both biotic treatment (e.g., afforestation

and grass-planting) and engineering measures (e.g., construction of terrace and check-dam
and gully control projects). Afforestation, grass-planting and construction of terrace are the
slope measures, while building of check-dams and gully control projects are the measures
on the channel. Although the utilized area of engineering measures was much smaller than
the biotic treatments, they can immediately and substantially trap streamflow and sediment
load. The fraction of the treated area (area treated by erosion control measures relative to





total catchment area) within the CSHC increased from 3.95% in the 1960s to 28.61% in the
2000s (Fig. 3b). The increase of the treated area was greatest during the 1980s as a result of
comprehensive management of small watersheds and during the 2000s due to the GFG
project since 1999. Some decreases in these areas occurred during the 1990s as some of the
erosion control measures undertaken were then subsequently destroyed.

For the fourteen sub-catchments, vegetation cover increased from 29.19 ± 21.09% in

1978 to 31.69 ± 17.18% in 1998, and then increased sharply to 44.10 ± 14.62% in 2010. In
the whole CSHC region, the amounts of vegetation cover in 1978, 1998 and 2010 were
23.61%, 25.68% and 38.71%, respectively. The increase of vegetation cover for the
catchments in the northwestern part (48.95% from 1978 to 1998 and 65.98% from 1998 to
2010) was greater than that in the southeastern part (-1.48% from 1978 to1998 and 28.72%
from 1998 to 2010). The annual LAI of the fourteen sub-catchments increased by 15.50%
from 1982-1999 to 2000-2011, and the relative change in the catchments of northwestern
part is 27.08%, which is greater than that in the southeastern part (6.82%). For the whole
CSHC region, the annual LAI changed from 0.51 during 1982-1999 to 0.55 during
2000-2011, an increase of 7.31%.
**3.2 Trends of hydro-meteorological and sediment yield variables**
Table 2 shows the trends in annual $P$, $Q$, $SSY$, $SC$ and $C_s$ of the fifteen catchments during
1961-2011. The annual $P$ showed a decline trend in all catchments but only significant in the
Xinshui and Zhouchuan catchments ($p<0.05$). The annual $Q$, $SSY$, $SC$ and $C_s$ showed
significant decreasing trends in all the catchments, and most of the decreases were at the
0.001 significance level. For the fourteen sub-catchments, the average decrease rates of





annual values of $Q$, $SSY$, $SC$ and $C_s$ were 0.86 mm yr$^{-1}$ (0.24-1.66 mm yr$^{-1}$), 190.06 t km$^{-2}$
yr$^{-1}$ (26.47-398.82 t km$^{-2}$ yr$^{-1}$), 2.73 kg m$^{-3}$ yr$^{-1}$ (0.69-4.70 kg m$^{-3}$ yr$^{-1}$) and 0.38 t km$^{-2}$ mm$^{-1}$
yr$^{-1}$ (0.04-0.87 t km$^{-2}$ mm$^{-1}$ yr$^{-1}$), respectively. For the whole CSHC region, the
corresponding change rates of $Q$, $SSY$, $SC$ and $C_s$ were -0.85 mm yr$^{-1}$, -131.52 t km$^{-2}$ yr$^{-1}$,
-2.06 kg m$^{-3}$ yr$^{-1}$ and -0.27 t km$^{-2}$ mm$^{-1}$ yr$^{-1}$, respectively. The annual average reductions in
the whole CSHC region are equivalent to 2.56%, 3.30%, 2.01% and 3.07% of the mean
annual values of $Q$, $SSY$, $SC$ and $C_s$, respectively.

The mean and the coefficient of variation, $C_v$, representing inter-annual variability of

annual values of $P$, $Q$, $SSY$, $SC$ and $C_s$ of the fifteen catchments during the three phases
(reference period-1, period-2 and period-3) are shown in Fig. 4. Compared to the reference
period, the mean annual precipitation decreased by 11.73% (6.36-15.69%) and 10.64%
(5.88-16.7%) on average in period-2 and period-3, respectively. From period-2 to period-3,
the change of mean annual precipitation was slight (increased by 1.32% on average) with
decrease of 2.45%-5.87% in four catchments and increase in remaining catchments
(0.35%-8.29%). The variability of annual $P$ also decreased as indicated by the reductions of
$C_v$ values during period-2 and period-3 (Fig. 4a). In contrast to annual $P$, the reductions of
mean annual $Q$, $SSY$, $SC$ and $C_s$ were clearly more evident. With respect to the reference
period, the reduction was 34.41% (9.45%-54.72%), 48.02% (17.98%-67.61%), 24.20%
(-9.93%-47.77%) and 39.31% (4.64%-63.5%) for $Q$, $SSY$, $SC$ and $C_s$ during period-2, and the
decreasing rate was even more in period-3 with values of 64.82% (36.72%-84.19%), 88.23%
(64.94% -97.64%), 67.81% (17.28%-91.12%) and 85.85% (63.51%-96.97%), respectively. $C_v$
of annual $Q$ increased in eight catchments, with the remaining ones showing decreasing



trends (Fig. 4b), $C_v$ values for $SSY$, $SC$ and $C_s$ increased in all catchments (Figs 4c-4e).
**3.3 Quantitative attribution of sediment load decline**
The effects of precipitation change and LUCC on sediment yield reductions in period-2 and
period-3 were quantified using Eqs. (2-5) and the results are shown in Fig. 5. The analysis
showed that both decreased precipitation and increased area treated with erosion control
measures contributed to the observed sediment load reduction, and that LUCC played the
major role. On average, the LUCC and precipitation change contributed 71.01% and
28.99%, respectively, to sediment load reduction from the reference period to period-2, and
their contributions were, respectively, 84.77% and 15.23% to sediment load reduction from
the reference period to period-3, respectively. The effect of LUCC in period-3 was greater
than that in period-2 as the land use and vegetation coverage had undergone substantial
changes due to the ecological restoration campaigns launched during period-3 (see Fig. 3).
From period-2 to period-3, the contribution of precipitation was negative for sediment yield
reduction in eleven catchments where the annual precipitation slightly increased during
these two periods, and thus the contribution of LUCC was larger than 100% (Fig. 5c). In
the remaining four catchments, the average contribution of LUCC increased to 86.15%.

In broad terms there are two factors that govern annual sediment yield of a catchment:

precipitation and landscape properties (soil, topography and vegetation). Higher
precipitation means higher streamflow, which is the immediate driver of erosion and
sediment transport. Landscape properties not only have an impact on the volume or
intensity of streamflow, but also determine the erodibility of the soil. On the basis of the
field evidence, we can hypothesize that the annual sediment yield $SSY$ can be expressed as





a product of a spatially variable component, which is only a function of a spatially variable
annual precipitation, $P$, and a temporally variable component, which is only a function of a
temporally variable fraction of area treated with erosion control measures, $A_c$.
$$SSY(\boldsymbol{x},t) = SSY_0 \cdot f_1[P(\boldsymbol{x})] \cdot f_2[A_c(t)] \qquad (6)$$
where $SSY_0$ is the sediment yield in the reference period, $t$ represents time and $\boldsymbol{x}$ is a vector
that represents spatial location of the catchments, and $f_1$ and $f_2$ are appropriate (yet to be
determined) functional forms that reflect the net effects of sub-catchment scale and
sub-annual runoff on sediment generation processes. In this framework, the variation of
$SSY$ during the reference period mainly depends on precipitation, and any spatial patterns
of $SSY$ among catchments may be controlled by differences in annual precipitation and land
surface conditions before LUCC took effect. As LUCC increased and took effect, the
temporal changes of $SSY$ may depend more on the fraction of treated surface area and
precipitation possibly might play a secondary role. Guided by this hypothesis, we next
organize the data analysis to generate separate spatial and temporal patterns that constitute
the respective components of the spatio-temporal patterns.
**3.4 Spatial pattern of the impacts of precipitation on sediment yield during Period 1**
The regression relationships between annual precipitation and sediment yield during the
reference period are shown in Fig. 6. Most of the catchments showed strong linear
correlation between precipitation and sediment yield. The coefficient of determination ($R^2$)
ranged from 0.24 to 0.85, and the correlation was significant in eight catchments ($p<0.05$)
(Table 3). Furthermore, the precipitation-sediment load relationship varied from catchment
to catchment and showed a spatial pattern. The correlation coefficient between precipitation





and sediment yield was greater for catchments in the northwestern part with average $R^2$
value of 0.69 and $p$ value of 0.017 compared to those in the southeastern part where the
average $R^2$ and $p$ values were 0.36 and 0.118, respectively (Table 3). Based on the slopes of
the regression equations between annual precipitation and sediment yield, the fifteen
catchments were classified into four groups, which indicate that the sediment production
capability of annual precipitation is different among the catchments (Fig. 6). The three
catchments of the first group in Fig. 6a had the greatest slopes (85-95) and the Shiwang
catchment in Fig. 6d had the lowest slope of 7.26. The average slope of the five catchments
in Group-2 was 57.57 (50-70), and the slope value of the six catchments in Group-3 was
30.88 (20-40). Overall, the regressed linear equations were significant for most of the
catchments, and were suitable for estimating the relative contributions of LUCC and
precipitation variability to sediment load changes.
Differences in catchment characteristics, including land use/cover, soil properties and
topography, as well as precipitation characteristics, are clearly the reason for the spatial
patterns in the precipitation-sediment yield relationship (Morera et al., 2013; Mutema et al.,
2015). To fully explore this, the mapping of information of catchment characteristics into
sediment yield models and simulating under climate scenarios is needed (Ma et al., 2014;
Achete et al., 2015). In this context, the inter-annual and intra-annual patterns of variability
of precipitation, including the distribution of storm events, may also contribute to the
observed spatial patterns of precipitation-sediment yield relationship.
**3.5 Spatial pattern of precipitation impacts on sediment yield during Periods 2 and 3**
Precipitation is the primary driver of runoff and, therefore, directly influences the sediment





transport capacity of streamflow and sediment yield at the catchment scale. Compared to
the reference period, the correlation between precipitation and sediment yield during the
period-2 decreased in the catchments, as indicated by the reductions of $R^2$ value in Table 3
and the increased scatter of the linear relationship in Fig. 7. The slope of the regression line
in the period-2 decreased in most of the catchments with respect to the reference period,
but in some catchments (e.g., Huangfu, Gushan and Kuye) the reduction in the slope was
slight. Furthermore, the precipitation-sediment yield relationship during these two periods
showed a similar spatial pattern. In period-2, the fifteen studied catchments could also be
classified into four groups using the decrease of slope of the regression line from Group-1
to Group-4 (Fig. 7). From the reference period to period-2, only Jialu catchment moved
from Group-1 to Group-2 and Yanhe catchment moved from Group-2 to Group-3 (Figs. 6
and 7).
In period-3, the correlation between precipitation and sediment yield was much weaker
compared to that during the reference period and period-2 (Table 3 and Fig. 8). The
relationship between precipitation and sediment yield was non-significant in all the
catchments (Table 3), and the scatter of the data points in Fig. 8 was notable. The slope of
the regression line during period-3 decreased sharply (Table 3), and for some catchments
the slope was even negative (Fig. 8d). This result indicates that the sediment production
capability of annual precipitation reduced greatly during period-3, and the increase of
precipitation amount in some catchments did not lead to an increase of sediment yield.
Furthermore, the spatial pattern of precipitation-sediment relationship during period-3 was
much different from that during the reference period and during period-2 through



comparisons of Fig. 8 with Figs. 6-7. As shown in Fig. 8, most of the catchments were
distributed in Group-1 and Group-4, which displayed considerable variability in the
precipitation-sediment relationship among the catchments.
The aforementioned analysis of precipitation-sediment yield relationship in different
periods clearly indicates that the impacts of precipitation on sediment load declined with
time, and the impacts were different among catchments, with a clear spatial pattern. The
decreased effects of precipitation on sediment load were consistent with the significant
reductions of sediment coefficient (Table 2) and the decreased contribution of precipitation
to sediment load reduction (28.99% and 15.23% in period-2 and period-3, respectively).
During period-2, the LUCC were mainly induced by SWCM, especially engineering
measures. During period-3, the combined effects of substantial vegetation cover and
conservation measures undoubtedly further weakened the effects of precipitation on
sediment load reduction.
As LUCC took effect during period-2 and period-3, and despite the much reduced role
of precipitation in driving changes in sediment yield, within-year temporal rainfall patterns
did play an important role in the observed changes of sediment yield, given that most of the
sediment yield was produced during a few key storm events. Taking the Yanhe catchment
as an example, the precipitation amount during the rainy season (May-October when
sediment load was measured) in 2003 and 2004 was 514.31 mm and 389.05 mm,
respectively, whereas the sediment load in 2004 ($2427.37 \times 10^4$ t) was about over four times
of that in 2004 ($590.04 \times 10^4$ t). As shown in Fig. 9, there were six days with precipitation
amount over 20 mm and the maximum daily precipitation amount on 25th Aug was 27.85



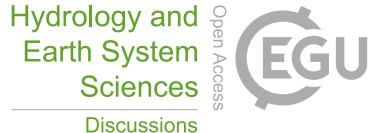
mm in 2003, and the values in 2004 were five days and 46.34 mm on 10th Aug.
Furthermore, heavy rainfall events were distributed in every month in 2003, whereas they
were concentrated in July and August in 2004. There were five evident peaks of sediment
load with the sum of $1646.24 \times 10^4$ t (67.82% of annual total) in 2004, especially the one on
10th Aug produced $784.53 \times 10^4$ t sediment load (32.32% of annual total) (Fig. 9b). In
contrast, there were three peaks of sediment load in 2003, and the maximum value was
only $139.97 \times 10^4$ t (Fig. 9a). Therefore, apart from annual precipitation amounts,
within-year rainfall patterns should also be considered to investigate the effects of
precipitation on temporal-spatial changes of streamflow and sediment load.
**3.6 Spatial pattern of the impacts of land use/cover on sediment load change**
The sediment load reductions in the LP were primarily caused by the LUCC and the
implementation of SWCM. The cropland area decreased 6233.13 km$^2$ (5.54% of region area)
and the forestland area increased 7246.45 km$^2$ (6.44% of region area) from 1986 to 2010.
Most of the increase in forestland area was converted from cropland area induced by the GFG
or reforestation project. As a result of the land use change, vegetation cover increased greatly
and it substantially contributed to the decreases of runoff and sediment production. The
SWCM, such as afforestation and engineering measures were the major interventions in the
study area to retain precipitation and consequently reduce streamflow and sediment load.
Establishing perennial vegetation cover was considered as one of the most effective measures
to stabilize soils and minimize erosion (Farley et al., 2005; Liu et al., 2014). It was reported
that both runoff coefficient and sediment concentration of catchments in the LP decreased
significantly and linearly with the vegetation cover (Wang et al., 2016). The engineering





structures mainly included creation of terrace and building of check-dams and reservoirs,
which reduced flood peaks and stored water and sediment within the catchment. There were
about 110, 000 check-dams in the LP which trapped about 21 billion m$^3$ of sediment during
the past six decades (Zhao et al., 2016). Over time, the effectiveness of engineering measures
decreased as they progressively fill with sediments, and vegetation restoration must in future
play a greater role in control of soil erosion for the LP.

To quantify the effects of SWCM on sediment load reduction, the relationship between

the sediment coefficient and the fraction of area treated with erosion control measures in the
15 catchments was analysed and the results are presented in Fig. 10. The sediment coefficient
decreased linearly with the fraction of treated land surface area in all catchments. The
correlation was significant in eleven catchments ($p<0.05$) with $R^2$ ranging from 0.78 to 0.99
(Table 4). Note that the temporal variations presented in Figure 10 are not based on annual
data (such data does not exist), but longer-term (decadal) averages.

The effects of SWCM on sediment load change show a spatial pattern. The correlation

between sediment coefficient and conservation measures was stronger in catchments located
in the north-western part compared to that in the south-eastern part (Table 4). Based on the
slope of the regression equation between the sediment coefficient and fraction of the treated
area, the 15 catchments were classified into three groups (Fig. 10), which indicated that the
degree of sediment load impacted by conservation measures was different among the
catchments. The four catchments of the first group in Fig. 10a had the greatest slopes over
0.85, followed by the four catchments in Group-2 (0.50~0.65) and seven catchments in
Group-3 (less than 0.30). Finally, inspired by the hypothesis presented in Eq. 6 and on the



basis of the observed linear relationships, it is now plausible to construct an empirical
relationship between a decadal average of sediment yield and the combination of decadal
average precipitation, $\overline{P}$, and the area under land use/cover change, $A_c$, of the following
form:
$$SSY = k_0.\overline{P}.(1 - k_1 A_c) \qquad (7)$$
**4   Conclusions**
The LP has undergone major changes in land use/land cover over the last 50 years as part
of a concerted effort to cut back on soil erosion and land degradation and sediment yield of
rivers. These included terrace and check-dam construction, afforestation, and pasture
reestablishment. Over the same period the region has also experienced some reduction in
rainfall, although this is relatively insignificant. Through analyses of hydrological and
sediment transport data, this study has brought out the long-term decreasing trends in
sediment loads across fifteen large sub-catchments located in the region. The study was
particularly aimed at extracting spatio-temporal patterns of sediment yield and attributing
these patterns to the broad hydro-climatic and landscape controls.

Over the study period (1961-2011), the total area undergoing erosion control treatment

went up from only 4% to over 30%. This included to decrease of cropland by 20%, increase
of forestland of 60% over the 40 years (grasslands remained unchanged), and an increase in
water body area by 90% (through the building of reservoirs). Over the same period annual
precipitation decreased by not more than 10%. As a result of the erosion control measures,
over the entire 50-year period, there have been major reductions in streamflow (65%),
sediment yield (88%), sediment concentration (68%) and sediment efficiency, i.e., annual





sediment yield/annual precipitation (86%).
The observed data in the 15 study catchments also exhibits interesting spatio-temporal
patterns in sediment yield. The study attempted to separate the relative contributions of
annual precipitation and LUCC to these spatio-temporal patterns. Before LUCC took effect
the data indicates a linear relationship between annual sediment yield and annual
precipitation in all 15 catchments, with highly variable slopes of the relationship between
the catchments, which exhibited systematic spatial patterns, in spite of considerable scatter.
As LUCC increased and took effect, the scatter increased and the slopes of the sediment
yield vs precipitation relationship became highly variable and lost any predictive power.
The study then looked at the controls on sediment coefficient instead of sediment yield
(thus eliminating the effect of precipitation and enabling a direct focus on landscape
controls). The results of this analysis found that sediment coefficient was heavily
dependent on the area under land use/cover treatment, exhibiting a linear (decreasing)
relationship. Even here, there was a considerable variation in the slope of the relationship
between the 15 catchments, which exhibited a systematic spatial pattern.
Preliminary analyses presented in this study suggests that much of the sediment yield in
the LP may be caused during only a few major storms. Therefore, the seasonality and
intra-annual variability of precipitation may play important roles in annual sediment yield,
which may also explain the spatial patterns of sediment yield and the effects of the various
LUCC. Also, the precipitation threshold for producing sediment yield would have increased
greatly as a result of SWCM and vegetation restoration in the LP. Exploration of these
questions in detail will require a more physically based model that can account for fine





scale rainfall variability. This is the next immediate step in our investigations, and will be
reported on in the near future.

**Acknowledgements**
This research was funded by the National Natural Science Foundation of China (41390464
and 41471094), the Chinese Academy of Sciences (GJHZ 1502) and the Youth Innovation
Promotion Association CAS (2016040). We thank the Ecological Environment Database of
Loess Plateau, the Yellow River Conservancy Commission, and the National Meteorological
Information Center for providing the hydrological and meteorological data. We thank Jianjun
Zhang for help in doing some figures and Zheng Ning for collecting the hydrological data.

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





**Figure captions**

**Figure 1.** Location of the studied catchments in the Coarse Sandy Hilly Catchments

(CSHC) region within the Loess Plateau.

**Figure 2.** Annual precipitation, streamflow and sediment load for the whole CSHC region

during 1961-2011.

**Figure 3.** The changes of (a) land use, (b) soil and water conservation measures area, (c)

vegetation cover and (d) LAI in the study area.

**Figure 4.** The changes of (a) precipitation, (b) streamflow, (c) sediment yield, (d) sediment

concentration and (e) sediment coefficient during different stages (1961-1969,

1970-1999 and 2000-2011).

**Figure 5.** Contributions of precipitation and land use/cover to reductions of sediment load

from (a) reference period (P1) to period-2 (P2), (b) reference period (P1) to period-3 (P3)

and (c) period-2 (P2) to period-3 (P3).

**Figure 6.** The relationship between annual precipitation and sediment yield during the

reference period (1961-1969).

**Figure 7.** The relationship between annual precipitation and sediment yield during the

period-2 (1970-1999).

**Figure 8.** The relationship between annual sediment yield and precipitation during the

period-3 (2000-2011).

**Figure 9.** Daily precipitation and sediment load of the Yanhe catchment during rainy

season (May-October) in (a) 2003 and (b) 2004.

**Figure 10.** Relationships between the sediment coefficient and percentage of the area

affected by soil and water conservation measures in the catchments. The data points

represent the average values of 1960s, 1970s, 1980s, 1990s, and 2000s.





**Table 1.** Long-term hydrometeorological characteristics (1961-2011) and leaf area index (LAI) (1982-2011) of the studied catchments in the Loess Plateau.

| ID | Catchment | Gauging station | Area ($km^2$) | Annual average | | | | | |
| | | | | $P$ (mm) | $Q$ (mm) | $SSY$ (t $km^{-2}$) | $SC$ (kg $m^{-3}$) | $C_s$ (t $km^{-2}$ $mm^{-1}$) | LAI |
|---|---|---|---|---|---|---|---|---|---|
| 1 | Huangfu | Huangfu | 3175 | 388.95 | 36.34 | 11608.86 | 275.90 | 27.35 | 0.351 |
| 2 | Gushan | Gaoshiya | 1263 | 422.49 | 49.55 | 12398.68 | 189.57 | 25.98 | 0.364 |
| 3 | Kuye | Wenjiachuan | 8515 | 394.63 | 59.25 | 9099.60 | 114.99 | 21.17 | 0.324 |
| 4 | Tuwei | Gaojiachuan | 3253 | 402.82 | 97.53 | 4454.47 | 38.44 | 10.16 | 0.363 |
| 5 | Jialu | Shenjiawan | 1121 | 445.51 | 49.22 | 9645.19 | 142.19 | 20.03 | 0.365 |
| 6 | Wuding | Baijiachuan | 29662 | 384.32 | 36.39 | 3089.61 | 74.09 | 7.67 | 0.371 |
| 7 | Qingjian | Yanchuan | 3468 | 485.58 | 38.93 | 8747.17 | 190.57 | 17.35 | 0.905 |
| 8 | Yanhe | Ganguyi | 5891 | 516.09 | 34.08 | 6604.90 | 166.31 | 12.45 | 1.623 |
| 9 | Shiwang | Dacun | 2141 | 572.16 | 32.99 | 798.89 | 20.32 | 1.31 | 2.158 |
| 10 | Qiushui | Linjiaping | 1873 | 469.02 | 34.83 | 7818.21 | 185.79 | 15.75 | 0.632 |
| 11 | Sanchuan | Houdacheng | 4102 | 486.23 | 50.37 | 3444.56 | 53.39 | 6.63 | 1.242 |
| 12 | Quchan | Peigou | 1023 | 539.73 | 30.24 | 7492.57 | 192.01 | 13.68 | 0.622 |
| 13 | Xinshui | Daning | 3992 | 529.96 | 29.22 | 3004.96 | 86.81 | 5.23 | 1.141 |
| 14 | Zhouchuan | Jixian | 436 | 530.06 | 30.13 | 4951.15 | 107.99 | 8.55 | 0.774 |
| 15 | CSHC | Toudaoguai and Longmen | 129654 | 437.27 | 33.30 | 3988.04 | 102.42 | 8.73 | 0.523 |



**Table 2.** Mann-Kendall trend analysis results for the annual precipitation ($P$), streamflow ($Q$), specific sediment yield ($SSY$), sediment concentration ($SC$), sediment coefficient ($C_s$) during 1961-2011.

| Catchment | $P$ | | $Q$ | | $SSY$ | | $SC$ | | $C_s$ | |
| --- | --- | --- | --- | --- | --- | --- | --- | --- | --- | --- |
| | $\beta$ (mm yr$^{-1}$) | $Z$ | $\beta$ (mm yr$^{-1}$) | $Z$ | $\beta$ (t km$^{-2}$ yr$^{-1}$) | $Z$ | $\beta$ (kg m$^{-3}$ yr$^{-1}$) | $Z$ | $\beta$ (t km$^{-2}$ mm$^{-1}$ yr$^{-1}$) | $Z$ |
| Huangfu | -0.52 | -0.57[ns] | -0.99 | -4.82*** | -323.24 | -4.50*** | -2.58 | -1.97* | -0.80 | -4.71*** |
| Gushan | -1.16 | -0.78[ns] | -1.47 | -5.02*** | -398.82 | -4.90*** | -3.92 | -3.75*** | -0.87 | -5.15*** |
| Kuye | -0.37 | -0.49[ns] | -1.66 | -5.98*** | -288.83 | -5.41*** | -3.22 | -4.61*** | -0.63 | -5.60*** |
| Tuwei | -0.27 | -0.24[ns] | -1.57 | -7.88*** | -130.34 | -5.20*** | -0.98 | -4.37*** | -0.30 | -5.59*** |
| Jialu | 0.26 | 0.19[ns] | -1.42 | -7.55*** | -298.10 | -5.36*** | -3.89 | -3.80*** | -0.69 | -5.60*** |
| Wuding | -0.37 | -0.39[ns] | -0.54 | -6.60*** | -79.19 | -4.55*** | -1.35 | -3.33*** | -0.20 | -4.94*** |
| Qingjian | -0.56 | -0.73[ns] | -0.24 | -2.06* | -138.54 | -3.01** | -3.53 | -3.09** | -0.30 | -2.73** |
| Yanhe | -1.17 | -1.19[ns] | -0.34 | -3.22* | -115.18 | -3.36*** | -3.07 | -3.30*** | -0.22 | -3.10** |
| Shiwang | -1.50 | -1.20[ns] | -0.61 | -4.01*** | -26.47 | -6.26*** | -0.69 | -5.43*** | -0.04 | -6.12*** |
| Qiushui | -0.35 | -0.28[ns] | -0.97 | -5.80*** | -290.44 | -6.98*** | -4.00 | -5.00*** | -0.55 | -5.98*** |
| Sanchuan | -1.71 | -1.43[ns] | -0.96 | -6.09*** | -108.69 | -5.35*** | -1.60 | -5.13*** | -0.21 | -5.99*** |
| Quchan | -1.14 | -0.94[ns] | -0.42 | -3.23* | -173.16 | -3.65*** | -4.12 | -3.72*** | -0.29 | -3.46*** |
| Xinshui | -2.71 | -2.37* | -0.70 | -5.57*** | -106.30 | -5.92*** | -1.92 | -3.77*** | -0.19 | -5.60*** |
| Zhouchuan | -2.48 | -2.21* | -0.79 | -7.20*** | -183.49 | -5.86*** | -4.70 | -6.73*** | -0.35 | -7.12*** |
| CSHC | -0.55 | -0.67[ns] | -0.85 | -5.91*** | -131.52 | -5.70*** | -2.06 | -4.26*** | -0.27 | -5.67*** |

[a] ***, ** and * indicate the significance levels of 0.001, 0.01 and 0.05, respectively. ns indicates the significance levels exceeds 0.05.



**Table 3.** The linear regression equations between annual precipitation and sediment load during three stages (1961-1969, 1970-1999 and 2000-2011).

| ID | Catchment | Reference period (1961-1969) | | | Period-2 (1970-1999) | | | Period-3 (2000-2011) | | |
|---|---|---|---|---|---|---|---|---|---|---|
| | | Regression equation | $R^2$ | $p$ | Regression equation | $R^2$ | $p$ | Regression equation | $R^2$ | $p$ |
| 1 | Huangfu | $y = 85.10x - 9752.3$ | 0.72 | 0.004 | $y = 88.67x - 12773$ | 0.37 | 0.000 | $y = 10.61x - 1386.4$ | 0.11 | 0.296 |
| 2 | Gushan | $y = 93.16x - 11606$ | 0.85 | 0.000 | $y = 84.91x - 12876$ | 0.31 | 0.001 | $y = 6.79x - 441.92$ | 0.11 | 0.291 |
| 3 | Kuye | $y = 66.42x - 5353.1$ | 0.55 | 0.022 | $y = 57.86x - 7048.9$ | 0.28 | 0.002 | $y = 1.95x - 33.53$ | 0.06 | 0.435 |
| 4 | Tuwei | $y = 34.35x - 2830.4$ | 0.84 | 0.001 | $y = 23.90x - 2913.2$ | 0.16 | 0.031 | $y = -2.05x + 1076.9$ | 0.05 | 0.469 |
| 5 | Jialu | $y = 90.23x - 7626.1$ | 0.75 | 0.005 | $y = 47.31x - 6155.7$ | 0.08 | 0.128 | $y = 9.98x - 1713.1$ | 0.07 | 0.405 |
| 6 | Wuding | $y = 21.76x - 649.01$ | 0.40 | 0.068 | $y = 16.96x - 1918.7$ | 0.24 | 0.006 | $y = 8.25x - 1291.3$ | 0.25 | 0.101 |
| 7 | Qingjian | $y = 49.75x - 5568.4$ | 0.27 | 0.154 | $y = 37.74x - 3904.1$ | 0.20 | 0.014 | $y = 8.15x - 580.79$ | 0.05 | 0.513 |
| 8 | Yanhe | $y = 56.63x - 11559$ | 0.34 | 0.100 | $y = 20.76x - 697.94$ | 0.13 | 0.053 | $y = 4.12x - 900.6$ | 0.05 | 0.831 |
| 9 | Shiwang | $y = 7.26x - 1255.3$ | 0.33 | 0.104 | $y = 5.02x - 1193$ | 0.22 | 0.008 | $y = -0.16x + 128.3$ | 0.02 | 0.667 |
| 10 | Qiushui | $y = 63.64x - 5673.8$ | 0.47 | 0.043 | $y = 42.20x - 6950$ | 0.22 | 0.008 | $y = -8.14x + 4488.1$ | 0.03 | 0.564 |
| 11 | Sanchuan | $y = 28.53x - 2109.1$ | 0.24 | 0.180 | $y = 18.75x - 3458.8$ | 0.36 | 0.000 | $y = -3.99x + 1901$ | 0.11 | 0.282 |
| 12 | Quchan | $y = 39.87x - 1265.6$ | 0.25 | 0.258 | $y = 20.51x + 60.83$ | 0.04 | 0.279 | $y = -26.78x + 13294$ | 0.11 | 0.300 |
| 13 | Xinshui | $y = 27.62x - 4923.5$ | 0.62 | 0.012 | $y = 19.27x - 4077.4$ | 0.48 | 0.000 | $y = 0.90x + 372.85$ | 0.01 | 0.777 |
| 14 | Zhouchuan | $y = 51.41x - 8690.4$ | 0.36 | 0.090 | $y = 42.27x - 10495$ | 0.28 | 0.003 | $y = -1.05x + 666.84$ | 0.07 | 0.408 |
| 15 | CSHC | $y = 33.13x - 4167.96$ | 0.60 | 0.015 | $y = 20.34x - 2728.2$ | 0.27 | 0.003 | $y = 2.03x + 174.09$ | 0.01 | 0.715 |





**Table 4.** Regression equations between the sediment coefficient and percentage of the area affected by soil and water conservation measures in the catchments.

| ID | Catchment | Regression equation | $R^2$ | $p$ |
|----|-----------|---------------------|-------|-----|
| 1 | Huangfu | $y = -0.67x+45.88$ | 0.85 | 0.025 |
| 2 | Gushan | $y = -0.90x+46.66$ | 0.82 | 0.034 |
| 3 | Kuye | $y = -0.83x+38.32$ | 0.89 | 0.017 |
| 4 | Tuwei | $y =-0.48x+19.94$ | 0.98 | 0.002 |
| 5 | Jialu | $y = -1.20x+53.20$ | 0.97 | 0.002 |
| 6 | Wuding | $y =-0.31x+16.92$ | 0.97 | 0.003 |
| 7 | Qingjian | $y = -0.31x+24.70$ | 0.48 | 0.193 |
| 8 | Yanhe | $y = -0.26x+18.54$ | 0.79 | 0.045 |
| 9 | Shiwang | $y = -0.15x+3.01$ | 0.87 | 0.020 |
| 10 | Qiushui | $y = -0.87x+35.69$ | 0.80 | 0.040 |
| 11 | Sanchuan | $y = -0.28x+13.32$ | 0.78 | 0.046 |
| 12 | Quchan | $y = -0.29x+21.02$ | 0.52 | 0.169 |
| 13 | Xinshui | $y = -0.20x+8.63$ | 0.72 | 0.069 |
| 14 | Zhouchuan | $y = -0.61x+17.89$ | 0.61 | 0.118 |
| 15 | CSHC | $y = -0.54x+17.74$ | 0.99 | 0.000 |





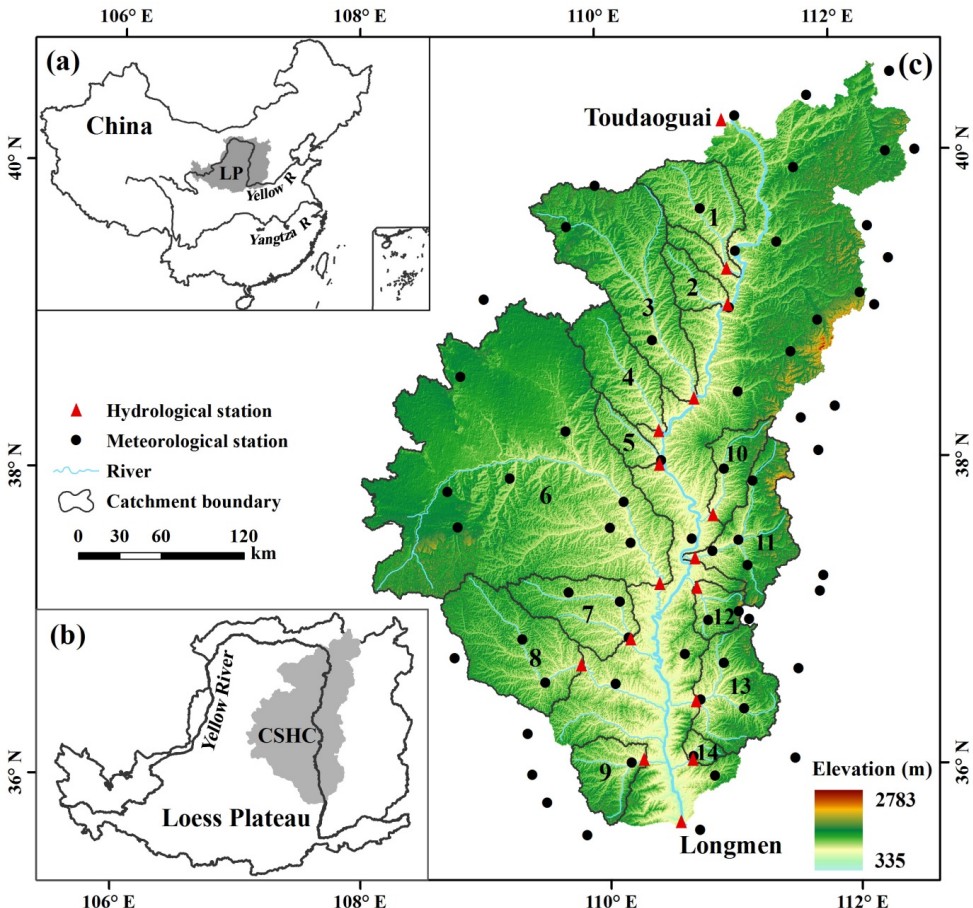

**Figure 1.** Location of the studied catchments in the Coarse Sandy Hilly Catchments (CSHC) region within the Loess Plateau.





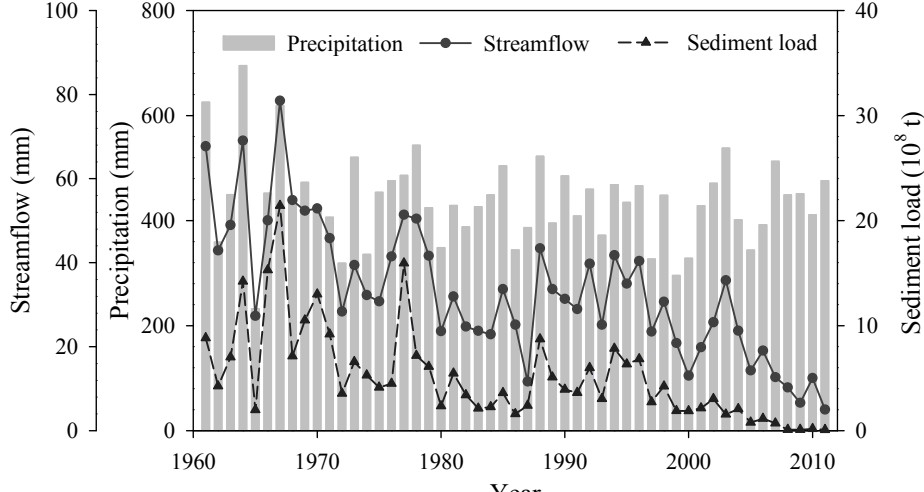

**Figure 2.** Annual precipitation, streamflow and sediment load for the whole CSHC region during 1961-2011.





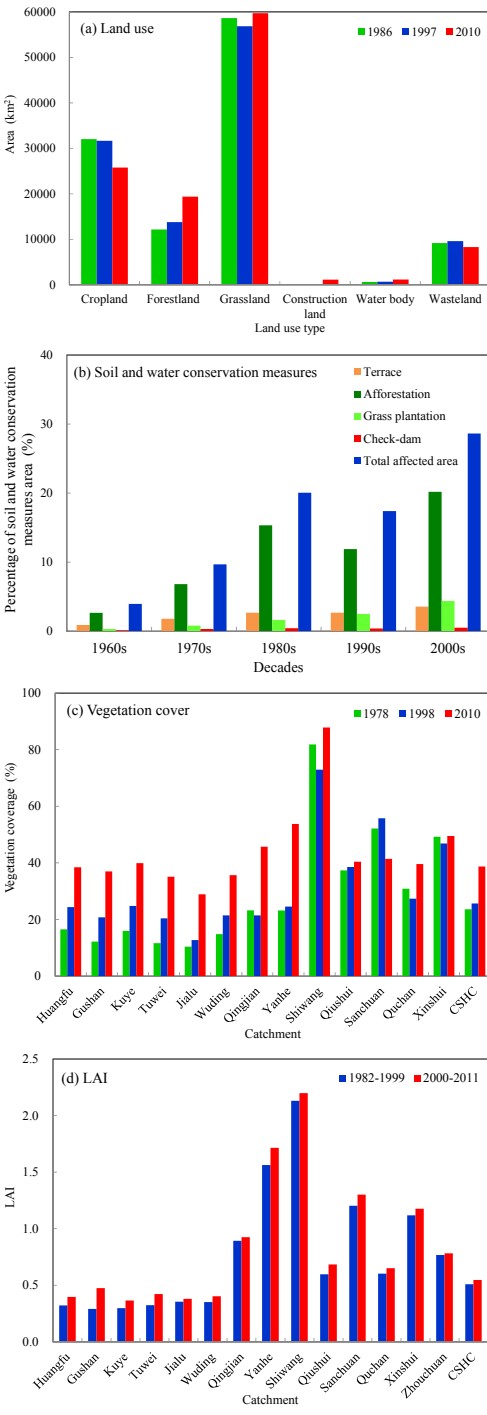

**Figure 3.** The changes of (a) land use, (b) soil and water conservation measures area, (c) vegetation cover and (d) LAI in the study area.






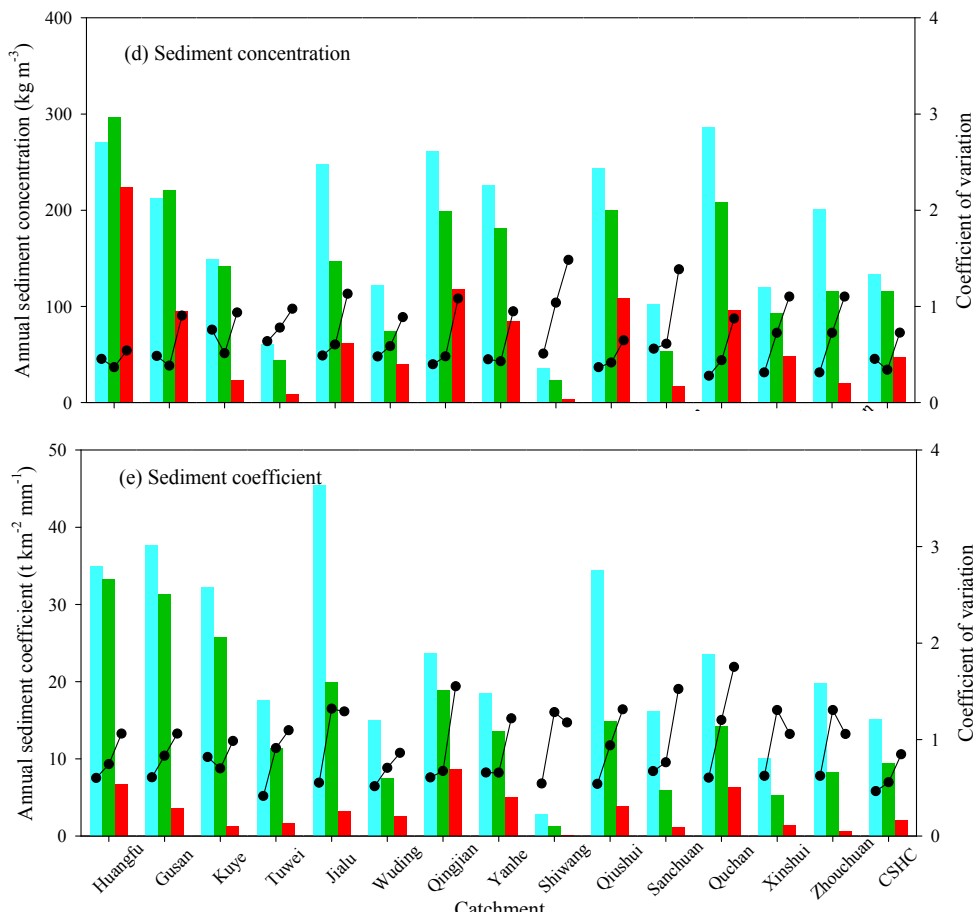

**Figure 4.** The changes of (a) precipitation, (b) streamflow, (c) sediment yield, (d) sediment concentration and (e) sediment coefficient during different stages (1961-1969, 1970-1999 and 2000-2011).





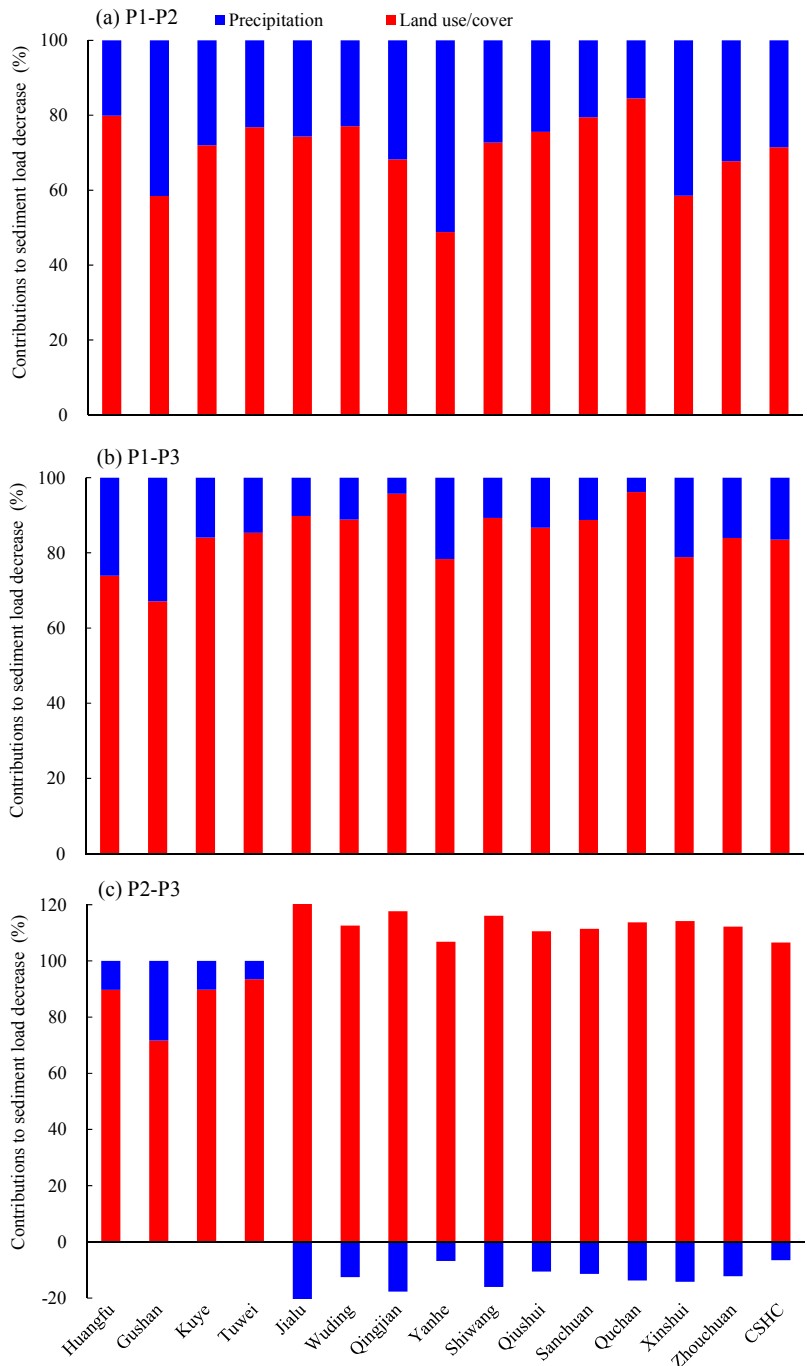

**Figure 5.** Contributions of precipitation and land use/cover to reductions of sediment load
from (a) reference period (P1) to period-2 (P2), (b) reference period (P1) to period-3 (P3) and
(c) period-2 (P2) to period-3 (P3).





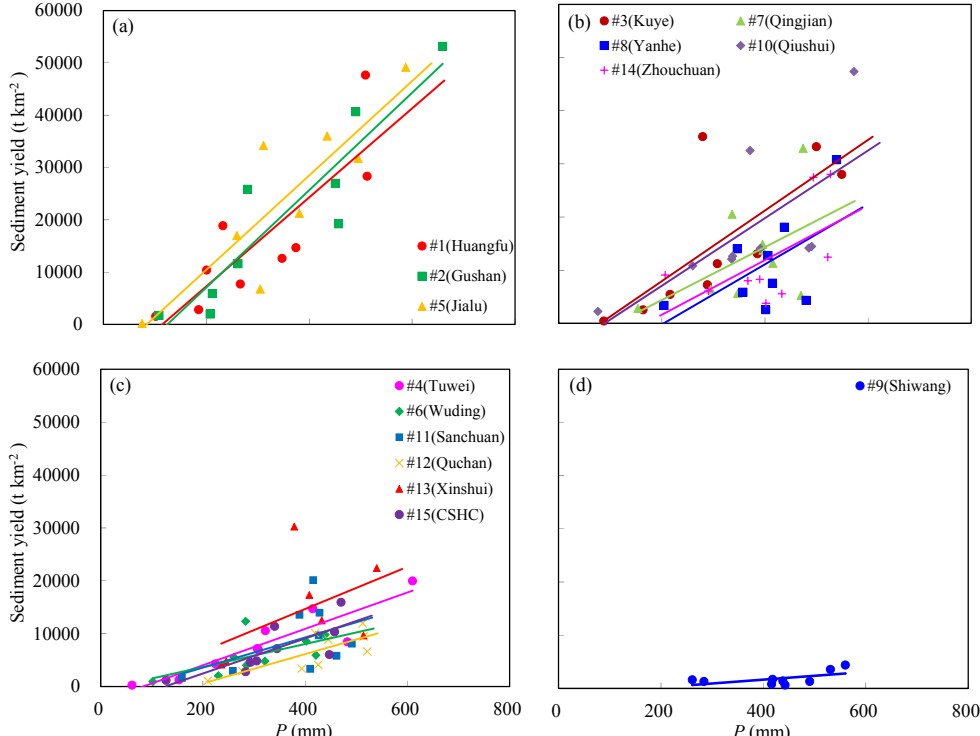

**Figure 6.** The relationship between annual precipitation and sediment yield during the reference period (1961-1969).





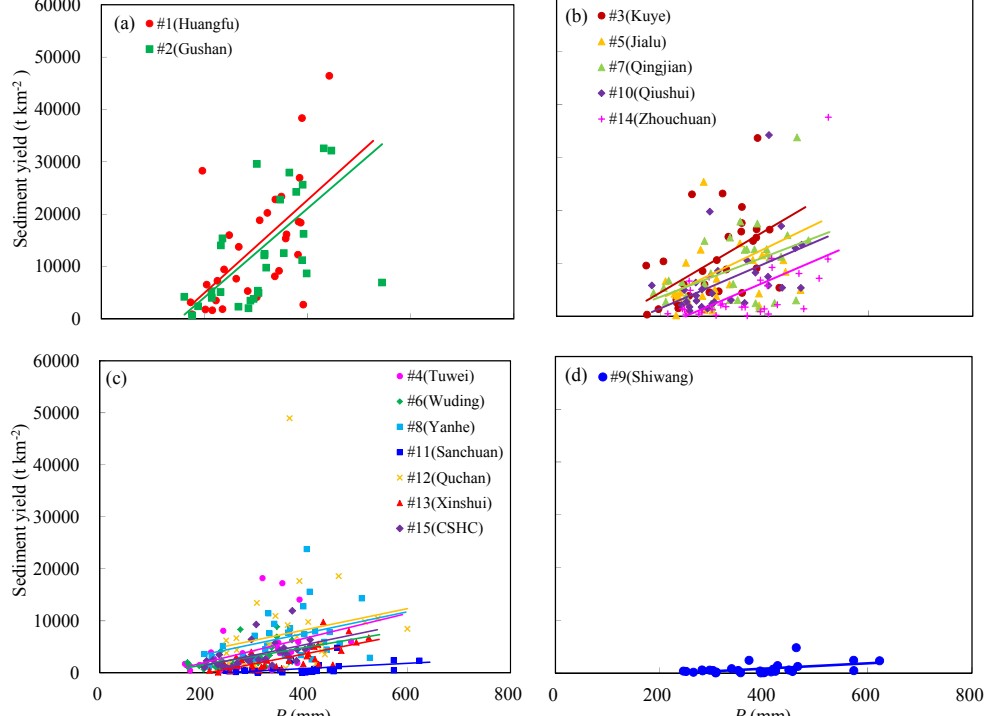

**Figure 7.** The relationship between annual precipitation and sediment yield during the

period-2 (1970-1999).





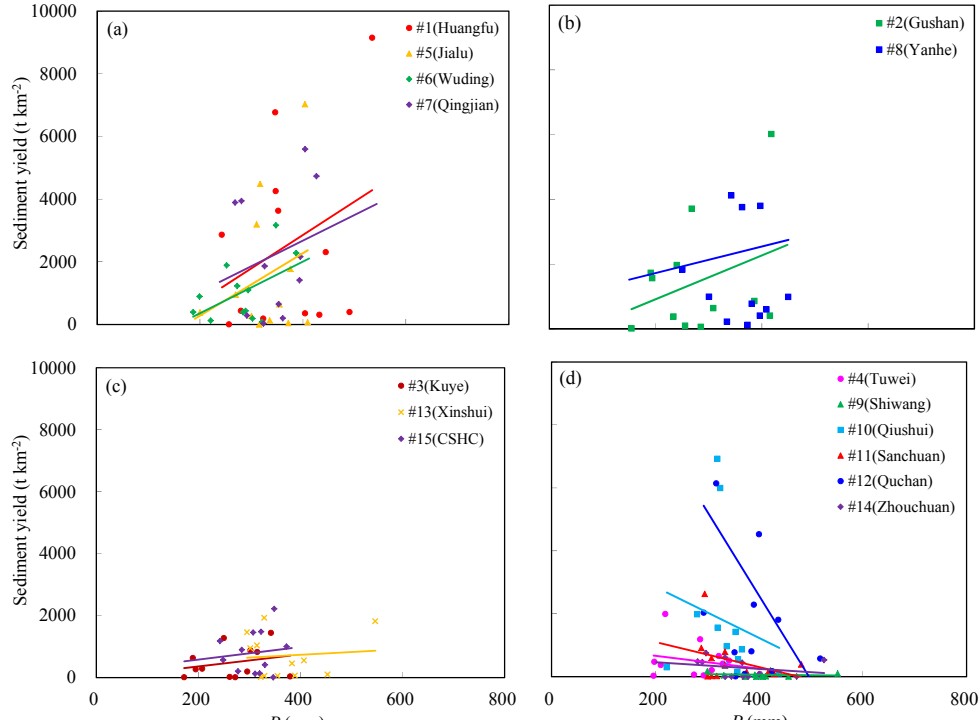

**Figure 8.** The relationship between annual sediment yield and precipitation during the

period-3 (2000-2011).





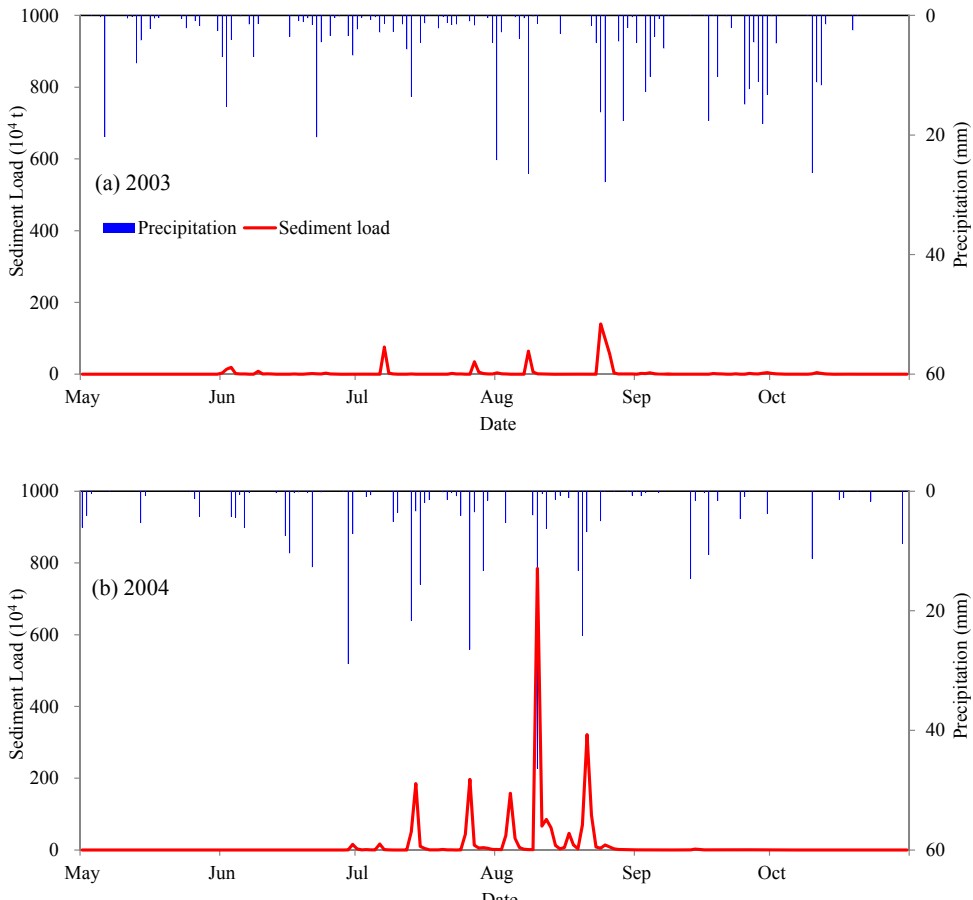

**Figure 9.** Daily precipitation and sediment load of the Yanhe catchment during rainy season

(May-October) in (a) 2003 and (b) 2004.



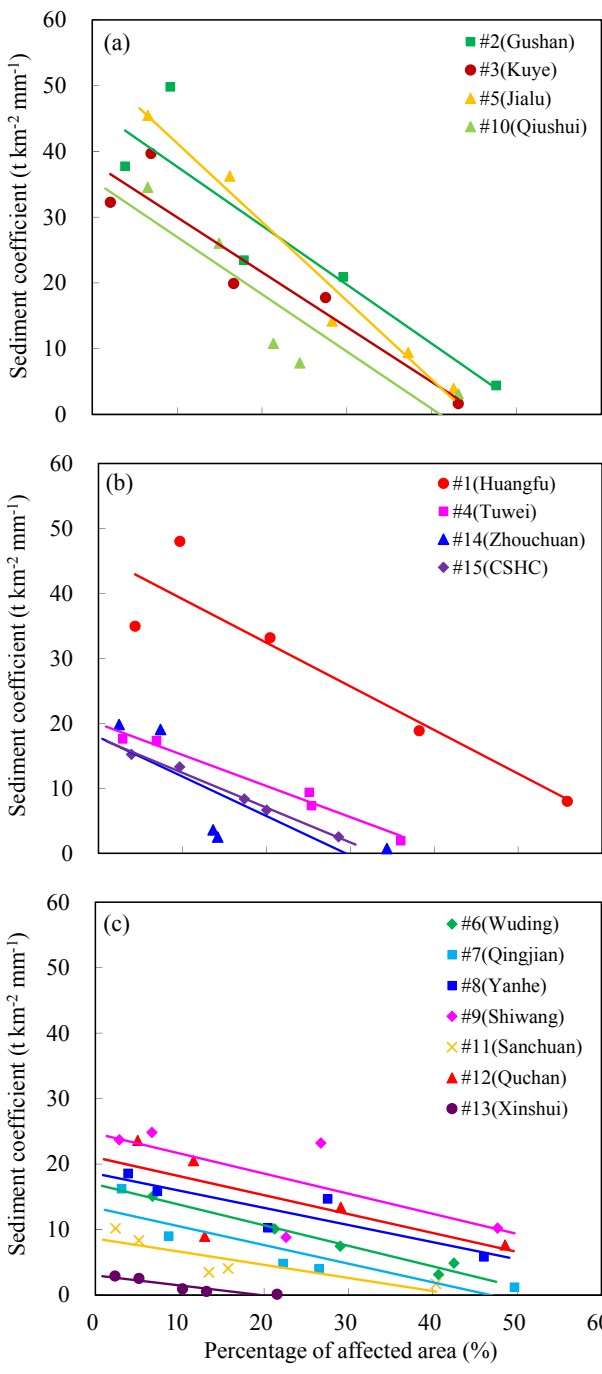

**Figure 10.** Relationships between the sediment coefficient and percentage of the area affected by soil and water conservation measures in the catchments. The data points represent the average values of 1960s, 1970s, 1980s, 1990s, and 2000s.