# Peer review of "Spatio-temporal patterns of the effects of precipitation variability"

_Hydrology and Earth System Sciences, 2016_

## Referee Comment (RC1) · Anonymous Referee #1 · 27 Feb 2017

The authors investigated the effects of precipitation variability and land use/cover changes (LUCC) on sediment yield in the Loess Plateau (LP), China. The author presents a detailed examination of the relationship between precipitation/LUCC and sediment yield in different catchments in the middle part of the LP during three periods. However, there are quite a few issues in this manuscript, hence I suggest some major revisions.

My major concerns are:

1. About the linear regression model for attribution analysis, nearly half of the catchments do not show statistically significant relationship between precipitation and sediment load during the reference period (Table 3). Therefore, it is very questionable to apply these linear regression models to the validation period for detecting the precipitation-induced (or LUCC-induced) sediment load change.

2. Even though this is just a "preliminary" study, as the author mentioned, I do not feel it is a complete work presented in this manuscript. There is a need for further discussion or analysis at some places. If the focus of this paper is on both spatial and temporal pattern of precipitation/LUCC-sediment relationships, there is lack of discussion on possible reasons for the spatial variability. Also, is it possible to investigate the effect of intra-annual variability of precipitation (or precipitation extremes) on sediment load since the authors have noted the effect is important (L328-330; L369-385)? Additionally, what are equation 6 and 7 for?

3. As the spatial pattern is the focus in section 3.4-3.6, I suggest to present the precipitation/LUCC-sediment relationships in maps rather than grouped scatter plots.

Specific comments:

P1, L22-23: Is the "70%" and "30%" a part of the conclusion in this study? If yes, I didn't see any of them in the results (section 3.3). Figure 5 does not support this statement either. If not, where are the numbers from? It would be better to also include it in the introduction.

P5, L106: The introduction above is mainly about the whole TP, why is only the middle part of LP investigated?

P6, L129: Any reference?

P7, L139-140: It would be better to describe the data first, then show the figure.

P7, L146-150: The whole sentence is a little bit confusing. SSY, SC, and Cs were estimated based on P, Q, and A?

[Figure]

P8, L158: What does vegetation cover mean? The vegetation fraction in each grid cell?

P13, L287-301: I am very confused that the authors proposed this "framework" but didn't show any results of it. What is its purpose here?

P15, L313-316: It would be better to describe the grouping at the beginning of this paragraph.

P17, L355-356: Does this indicate that the precipitation-sediment relationship gets stronger in some regions but weaker in some other regions? Is the strengthened (or weakened) relationship related to the SWCM or vegetation change in these catchments?

P20, L425: The same issue as (P13, L287-301). What is k0 and k1?

---

## Referee Comment (RC2) · Anonymous Referee #2 · 27 Feb 2017

The author's attempt to determine the drivers of changes in sediment yield within the Coarse Sandy Hill Catchments region of the Loess Plateau. The authors attribute changes in sediment yield to both land-use change and changes in precipitation. Although the authors do a great job characterizing changes in precipitation, land cover, and sediment yield, their statistical analysis leaves much room for improvement and many of their figures could be clarified.

While land-use change (specifically crop to forest) and precipitation change are certainly major drivers in changes in sediment yield, soil properties, topography, and

changes in urban cover must also play some role, and thus warrant some discussion as to their exclusions, or what excluding them might mean for the paper's results. Moreover, as the author's bring up, the intensity of certain storms are not always captured when one looks at annual average precipitation, but these intense storm greatly affect sediment yield. Thus, analyzing the number of intense events along with average precipitation may prove insightful.

Lines 35-36. The effect of precipitation is also temporally variable, yet it is framed in the abstract and throughout most of the paper as only being spatially variable.

Lines 144-145. Although the author's provided a robust motivation for their analysis of the their 14 chosen catchments within the CSHC, a sentence or two explaining why they are studying the CSHC would be useful.

Line 179. Why resample the AVHRR data?

Lines 179-185. What is meant by vegetation cover? Do the authors estimate vegetation cover using NDVI or a different vegetative index?

Lines 183-185. It seems as though the authors have useful spatial information regarding the total areas impacted by conservation measures (the Yao et al. 2011) dataset, yet it's unclear where this comes into play in their analysis.

Lines 192-194. Did you test your variables after performing the TFPW to see if any residual autocorrelation remained?

Lines 220-226. What was the land-cover during the study period which the authors consider their reference period where "the effects of human activities were slight and could be mostly ignored." Here and throughout, presenting the spatial data as maps rather than bar graphs or scatter plots will more clearly to the audience. Especially given in the results and discussion where the authors often reference the differences in spatial patterns.

Line 254. As mentioned above, need to be clear about the proxy used for vegetation

cover.

Line 260, 263. Average annual LAI?

Line 278-288. Why use the coefficient of variation and not standard deviation? What do these results tell us?

Section 3.3. In Equation 6, precipitation is also a temporally variable component, and 'area treated with erosion control measures' is also a spatially variable component. And it seems as though other factors (steeper slopes, soil properties, impermeable surface area, etc) may also play a role in affecting SSY. Moreover, it seems likely that changes in precipitation and land-use change may interact to affect sediment yield. The authors may want to rethink the way they've framed their analysis. Especially as 6/14 catchments in their analysis exhibited no significant correlation. A multiple regression analysis with an interaction term may be a more appropriate means of analysis.

Lines 387 and throughout: Authors often discuss a 'clear spatial pattern' present in their results, thus maps would be more useful as figures than scatter plots.

Line 393 and 419. Remove undoubtedly.

Lines 449-454. Not quite sure how this resulting empirical relationship follows from the preceding analysis. What are $k_0$ and $k_1$. Also, once better explained, the authors could prove this empirical relationship is robust by showing how accurately it predicts SSY when they input observational data.

Table 2: Add an ID column.

---

## Author Comment (AC2) · 8 May 2017

First of all, I greatly appreciate the time the reviewers put in reading my manuscript, and the comments provided were valuable and constructive. To be more comprehensive, I have provided a list of itemized responses to all review comments in the attached supplemental file (instead of responding to each reviewer's comments individually). Hopefully I have adequately addressed all the review comments. Please also note the Supplement to this comment.

Please also note the supplement to this comment:
http://www.hydrol-earth-syst-sci-discuss.net/hess-2016-654/hess-2016-654-AC2-
supplement.pdf

―――――――――――――――――――――

---

## Author Response (AR1)

May 30, 2017

Memorandum

To:        Pro. Nunzio Romano, Editor of *Hydrology and Earth System Sciences*

Subject:   **Revision of hess-2016-654**

Dear Pro. Nunzio Romano:

Upon your request, we have carefully addressed all the comments made by the two anonymous reviewers on our manuscript (hess-2016-654) entitled "Spatio-temporal patterns of the effects of precipitation variability and land use/cover changes on long-term changes in sediment yield in the Loess Plateau, China" and revised the manuscript accordingly. The comments have helped us greatly improve the overall quality of the manuscript. We added Jianjun Zhang, Yu Liu and Zheng Ning to the author list as a result of their important contributions to the revision of this paper. The following is the point-point response to all the comments. The page and line numbers in the following response refer to the revised manuscript with changes marked.

**Response to Anonymous Referee #1:**

General comments:

The authors investigated the effects of precipitation variability and land use/cover changes (LUCC) on sediment yield in the Loess Plateau (LP), China. The author presents a detailed examination of the relationship between precipitation/LUCC and sediment yield in different catchments in the middle part of the LP during three periods. However, there are quite a few issues in this manuscript, hence I suggest some major revisions.

Reply: *All the issues have been carefully considered and modifications have been done accordingly (see the following point-to-point replies to the comments).*

My major concerns are:

1. Comment:

About the linear regression model for attribution analysis, nearly half of the catchments do not show statistically significant relationship between precipitation and sediment load during the reference period (Table 3). Therefore, it is very questionable to apply these linear regression models to the validation period for detecting the precipitation-induced (or LUCC-induced) sediment load change.

Reply: *We have changed the linear regression model (SSY=aP+b) for attribution analysis (see P.10, Lines 289-293). Rustomji et al. (2008, Water Resources Research, 44, W00A04, doi:10.1029/2007WR006656) found that the square root of sediment yield in the catchments of the Loess Plateau was linearly related to the precipitation. This was used in this study as the motivation to develop the precipitation-sediment yield relationship during the reference period:*

$$\sqrt{SSY} = aP + b$$

(1)

*Within this new regression model, the sediment yield was correlated with precipitation at 0.05 level in eleven catchments and 0.1 level in three catchments, and the R² value also improved much compared to the original linear regression model (see Table 1). Therefore, this new regression model was satisfactory to detect the precipitation-induced (or LUCC-induced) sediment load change (see P.16, Lines 527-534).*

**Table 1.** The linear regression equations between square root of specific sediment yield and annual precipitation ($\sqrt{SSY} = aP + b$) during the reference period (1961-1969).

| ID | Catchment | Regression equation | $R^2$ | $p$ |
|----|-----------|---------------------|-------|-----|
| 1 | Huangfu | $y = 0.341x + 12.041$ | 0.78 | 0.002 |
| 2 | Gushan | $y = 0.349x + 8.237$ | 0.84 | 0.001 |
| 3 | Kuye | $y = 0.323x + 9.939$ | 0.67 | 0.007 |
| 4 | Tuwei | $y = 0.218x + 12.635$ | 0.87 | 0.000 |
| 5 | Jialu | $y = 0.382x + 6.976$ | 0.78 | 0.004 |
| 6 | Wuding | $y = 0.174x + 20.544$ | 0.53 | 0.027 |
| 7 | Qingjian | $y = 0.232x + 20.923$ | 0.48 | 0.040 |
| 8 | Yanhe | $y = 0.243x + 0.741$ | 0.39 | 0.070 |
| 9 | Shiwang | $y = 0.070x + 10.935$ | 0.27 | 0.150 |
| 10 | Qiushui | $y = 0.257x + 30.738$ | 0.60 | 0.014 |
| 11 | Sanchuan | $y = 0.191x + 15.053$ | 0.36 | 0.089 |
| 12 | Quchan | $y = 0.202x + 34.590$ | 0.72 | 0.016 |
| 13 | Xinshui | $y = 0.202x - 6.593$ | 0.71 | 0.004 |
| 14 | Zhouchuan | $y = 0.207x + 20.226$ | 0.33 | 0.090 |
| 15 | CSHC | $y = 0.218x + 5.689$ | 0.70 | 0.005 |

2. Comment:

Even though this is just a "preliminary" study, as the author mentioned, I do not feel it is a complete work presented in this manuscript. There is a need for further discussion or analysis at some places. If the focus of this paper is on both spatial and temporal pattern of precipitation/LUCC-sediment relationships, there is lack of discussion on possible reasons for the spatial variability. Also, is it possible to investigate the effect of intra-annual variability of precipitation (or precipitation extremes) on sediment load since the authors have noted the effect is important (L328-330; L369-385)? Additionally, what are equation 6 and 7 for?

Reply: *This is a good comment.*

*First, we have showed the spatial distribution of catchment characteristics (including precipitation, soil, slope, LAI and LUCC), which are the possible reasons for the spatial variability of precipitation/LUCC-sediment relationships (see Figs. 1-3). The precipitation/LUCC-sediment relationships were also presented in maps to indicate spatial pattern rather than grouped scatter plots used in the original version (see Figs. 4-5).*

*Second, we have compared the precipitation/LUCC-sediment relationships in different parts of the study area, and discussed the effects of catchment characteristics on the variability of relationships among catchments (see P.16, Lines 535-545; P.18, Lines 710-711, 718-724; P.21, Lines 800-810). We have also investigated the effects of potential factors (precipitation, percentage area of forestland, grassland, construction land, terracing and check-dams, and LAI) on sediment yield change in different stages (see P.15, Lines 469-476). It was found that check-dam construction was the dominant factor for sediment yield reduction from reference period to period-2, and pasture plantation and check-dam construction acted the dominant factors for sediment yield from reference period to period-3 (see Table 2). The increase of precipitation mitigated the reduction of sediment yield to some degree from period-2 to period-3 (see Table 2).*

*Third, for the effect of intra-annual variability of precipitation (or precipitation extremes) on sediment load, we have investigated the correlation between sediment yield and storm events (including storm numbers, precipitation amount of storms) in the study area during different decades (see P.19, Lines 738-745). The analysis showed that the sediment yield was significantly correlated with storm numbers in the 1960s, 1970s and 1980s (p<0.05), and precipitation amount of storms in the 1960s and 1970s (p<0.05) (see Table 3). This result indicated the critical role of storm events in sediment yield, especially during the periods before substantial LUCC took effect. Furthermore, we have chosen a catchment as an example and compared the within-year rainfall pattern and sediment load in nearby two years (see Fig. 6). The comparison also indicated the important role of distribution of storm events in sediment yield (see P.19, Line 746 to P.20, Line 764).*

*Fourth, we have deleted equations 6 and 7 which are somewhat misleading, and reframed the analysis about the spatio-temporal patterns of the impacts of precipitation and LUCC on sediment yield (see P.15, Line 477 to P.16, Line 525). We divided the study period into three stages including reference period and validation periods (period-2 and period-3). We used the maps to show the spatial distribution of precipitation-sediment relationships in the three stages and investigated the reason for the spatial-temporal variability (see Figs. 4-5). In the reference period before LUCC took effect, the variation of SSY mainly depends on precipitation, and any spatial patterns of SSY among catchments was controlled by differences in annual precipitation and land surface conditions. During the validation period (period-2 and period-3) with LUCC increased and took effect, the temporal changes of SSY depend more on the fraction of treated surface area and precipitation played a secondary role. The spatial pattern of the impacts of precipitation on sediment yield was dependent on the landscape properties among catchments, and it changed considerably especially in period-3 as the combined effects of engineering measures and vegetation restoration project.*

*Finally, it should be noted that the focus of this study is to present the spatio-temporal patterns of effects of precipitation variability and LUCC on long-term changes in sediment yield, and it is a "preliminary" study to investigate the detailed effects of intra-annual variability of precipitation and catchment characteristics on sediment yield, which need more detailed processes-based studies on fine scales. This is the next immediate step in our investigations (see P.18, Lines 722-727; P.23, Lines 884-892).*

[Figure]

**Figure 1.** Spatial distribution of (a) annual mean precipitation (1961-2011), (b) growing season leaf area index (LAI, 1982-2011), (c) soil type and (d) slope in the study area.

[Figure]

**Figure 2.** Land use and cover of the study area in (a) 1975, (b) 1990, (c) 2000 and (d) 2010.

[Figure]

**Figure 3.** Long-term trends in growing season LAI changes over (a) 1982-2011, (b) 1982-1999 and (c) 2000-2011 in the study area. Inset in each figure shows the frequency distribution of the LAI trends.

[Figure]

**Figure 4.** Spatial distribution of slope $a$ in the regression equation $\sqrt{SSY} = aP + b$ during (a) reference period (1961-1969), (b) period-2 (1970-1999) and (c) period-3 (2000-2011). $SSY$ is specific sediment yield, and $P$ is precipitation.

[Figure]

**Figure 5.** Spatial distribution of slope $m$ in the regression equation $\overline{SC} = -mA_c + n$. $\overline{SC}$ is the decadal average sediment coefficient, and $A_c$ is the fraction area treated with soil and water conservation measures.

**Table 2.** The regression models for sediment yield change ($\Delta SSY$) in different stages.

| Period | Regression model | $R^2$ | $p$ |
|---|---|---|---|
| Reference period vs. Period-2 | $\Delta SSY=-0.135-0.850 \times \Delta Dam$ | 0.886 | 0.000 |
| Reference period vs. Period-3 | $\Delta SSY=-0.067-0.659 \times \Delta Dam-0.081 \times \Delta Pasture$ | 0.928 | 0.023 |
| Period-2 vs. Period-3 | $\Delta SSY=-0.105-0.488 \times \Delta Dam+0.058 \times \Delta P-0.129 \times \Delta Pasture$ | 0.905 | 0.003 |

$\Delta Dam$ and $\Delta Pasture$ are changes in percentage area of check-dams and pasture plantation, respectively. $\Delta P$ is changes of annual precipitation over the two compared periods.

**Table 3.** Pearson correlation coefficients ($r$) and two-tailed significance test values ($p$) between sediment yield and annual precipitation ($P$), number of storms ($N_{storm}$) and precipitation amount of storms ($P_{storm}$) during different decades of the CSHC region.

| Decades | $P$ | | $N_{storm}$ | | $P_{storm}$ | |
|---|---|---|---|---|---|---|
| | $r$ | $p$ | $r$ | $p$ | $r$ | $p$ |
| 1960s | 0.772 | 0.015* | 0.808 | 0.008** | 0.718 | 0.029* |
| 1970s | 0.266 | 0.458 | 0.714 | 0.020* | 0.695 | 0.026* |
| 1980s | 0.775 | 0.009** | 0.633 | 0.050* | 0.527 | 0.117 |
| 1990s | 0.865 | 0.001*** | 0.591 | 0.072 | 0.572 | 0.084 |
| 2000s | 0.118 | 0.715 | 0.006 | 0.986 | 0.138 | 0.669 |

***, ** and * indicate the significance levels of 0.001, 0.01 and 0.05, respectively.

[Figure]

**Figure 6.** Daily precipitation and sediment load of the Yanhe catchment during rainy season (May-October) in (a) 2003 and (b) 2004.

3. Comment:

As the spatial pattern is the focus in section 3.4-3.6, I suggest to present the precipitation/LUCC-sediment relationships in maps rather than grouped scatter plots.

Reply: *This is a good suggestion. We have used maps to present the spatial pattern of the precipitation/LUCC-sediment relationships (see Figs 4-5).*

Specific comments:

1. Comment:

P1, L22-23: Is the "70%" and "30%" a part of the conclusion in this study? If yes, I didn't see any of them in the results (section 3.3). Figure 5 does not support this statement either. If not, where are the numbers from? It would be better to also include it in the introduction.

Reply: *The attribution analysis indicated that the contribution of LUCC to sediment load was 74.39% from the reference period to period-2, and it was 88.67% from the reference period to period-3 (see P.14, Lines 436-439). Therefore, it can be considered that "The human induced land use/cover change (LUCC) was the dominant factor with contributing over 70% of the sediment load reduction, whereas the contribution of precipitation was less than 30%" (see P.2, Lines 31-33).*

2. Comment:

P5, L106: The introduction above is mainly about the whole TP, why is only the middle part of LP investigated?

Reply: *Most of sediment yield of the LP was produced in the Coarse Sandy Hilly Catchments (CSHC) region in the central region of the LP, which supplied over 70% of total sediment load in the YR, especially coarse sand. The CSHC region was the focus of our efforts to investigate the variation of sediment load within the LP (see P.5, Line 130 to P.6, Line 137).*

3. Comment:

P6, L129: Any reference?

Reply: *These two percentages were determined with the observed hydrological data during 1961-2011 by us (see P.7, Lines 173-174).*

4. Comment:

P7, L139-140: It would be better to describe the data first, then show the figure.

Reply: *We have described the changes of annual precipitation, streamflow and sediment load of the CSHC region with the data first, and then showed the figure (see P.7, Lines 185-190).*

5. Comment:

P7, L146-150: The whole sentence is a little bit confusing. SSY, SC, and Cs were estimated based on P, Q, and A?

Reply: *We have rephrased this sentence. The SSY, SC and $C_s$ was estimated based on the observed P, Q, S and A (SSY=S/A, SC=S/(Q.A), $C_s$=SSY/P) (see P.8, Lines 208-212).*

6. Comment:

P8, L158: What does vegetation cover mean? The vegetation fraction in each grid cell?

Reply: *The LAI was used in this study to indicate vegetation cover (see P.8, Lines 218-220).*

7. Comment:

P13, L287-301: I am very confused that the authors proposed this "framework" but didn't show any results of it. What is its purpose here?

Reply: *We have reframed the analysis about the spatio-temporal patterns of precipitation and LUCC impacts on sediment yield. Please see the fourth point of the response to #2 main comment.*

8. Comment:

P15, L313-316: It would be better to describe the grouping at the beginning of this paragraph.

Reply: *We have described the spatial distribution and grouping of the precipitation-sediment yield relationships at the beginning of the paragraph (see P.16, Lines 539-545).*

9. Comment:

P17, L355-356: Does this indicate that the precipitation-sediment relationship gets stronger in some regions but weaker in some other regions? Is the strengthened (or weakened) relationship related to the SWCM or vegetation change in these catchments?

Reply: *We have deleted this misleading sentence. In period-3, as a result of the combined effects of SWCM and vegetation restoration project, the precipitation-sediment relationship became much weaker in all the catchments, and slope of the regression equation decreased sharply in all the catchments, especially for some catchments the slope was even negative (see Fig. 5). Furthermore, the spatial pattern of precipitation-sediment relationship had somewhat change compared to that in the reference period and period-2 (see P.17, Lines 658-670).*

10. Comment:

P20, L425: The same issue as (P13, L287-301). What is k0 and k1?

Reply: *We have deleted this misleading equation and the related statements.*

**Response to Anonymous Referee #2:**

1. Comment:

The author's attempt to determine the drivers of changes in sediment yield within the Coarse Sandy Hill Catchments region of the Loess Plateau. The authors attribute changes in sediment yield to both land-use change and changes in precipitation. Although the authors do a great job characterizing changes in precipitation, land cover, and sediment yield, their statistical analysis leaves much room for improvement and many of their figures could be clarified.

Reply: *We have improved the statistical analysis (see response to the main comment #1 of reviewer #1), clarified many figures in map not in bar graphs or scatter plots (see Figs. 1-5), and addressed all the following comments.*

2. Comment:

While land-use change (specifically crop to forest) and precipitation change are certainly major drivers in changes in sediment yield, soil properties, topography, and changes in urban cover must also play some role, and thus warrant some discussion as to their exclusions, or what excluding them might mean for the paper's results. Moreover, as the author's bring up, the intensity of certain storms are not always captured when one looks at annual average precipitation, but these intense storm greatly affect sediment yield. Thus, analyzing the number of intense events along with average precipitation may prove insightful.

Reply: *We have investigated the possible effects of catch characteristics (soil, slope, LUCC and LAI) on the changes of sediment yield (see the second point of the response to #2 main comment of reviewer*

*#1). We have also investigated the effects of storm events on sediment yield (see the third point of the response to #2 main comment of reviewer #1).*

3. Comment:

Lines 35-36. The effect of precipitation is also temporally variable, yet it is framed in the abstract and throughout most of the paper as only being spatially variable.

Reply: *Yes, the effect of precipitation was both temporally and spatially variable. We have reframed the analysis about the spatio-temporal patterns of precipitation impacts on sediment yield (see the fourth point of the response to #2 main comment of reviewer #1).*

4. Comment:

Lines 144-145. Although the author's provided a robust motivation for their analysis of the their 14 chosen catchments within the CSHC, a sentence or two explaining why they are studying the CSHC would be useful.

Reply: *We have explained the reason for focusing on the CSHC region in this study (see the response to #2 specific comment of reviewer #1).*

5. Comment:

Line 179. Why resample the AVHRR data?

Reply: *We have changed the data source of LAI data. LAI was derived from the Global Land Surface Satellite (GLASS) NDVI series with spatial resolution of 1 km (see P.8, Lines 218-220).*

6. Comment:

Lines 179-185. What is meant by vegetation cover? Do the authors estimate vegetation cover using NDVI or a different vegetative index?

Reply: *The LAI was used in this study to indicate vegetation cover (see P.8, Lines 218-220).*

7. Comment:

Lines 183-185. It seems as though the authors have useful spatial information regarding the total areas impacted by conservation measures (the Yao et al. 2011) dataset, yet it's unclear where this comes into play in their analysis.

Reply: *We only had the total area of conservation measures, not the spatial information (see P.8, Lines 220-222). It is necessary to obtained them with field survey and high-resolution satellite data, and investigate the detailed effects of conservation measures on streamflow and sediment yield.*

8. Comment:

Lines 192-194. Did you test your variables after performing the TFPW to see if any residual autocorrelation remained?

Reply: *There was no residual autocorrelation remaining after performing the TFPW (see P.9, Lines 257-258).*

9. Comment:

Lines 220-226. What was the land-cover during the study period which the authors consider their reference period where "the effects of human activities were slight and could be mostly ignored." Here and throughout, presenting the spatial data as maps rather than bar graphs or scatter plots will more clearly to the audience. Especially given in the results and discussion where the authors often reference the differences in spatial patterns.

Reply: *The land use and cover in 1975, 1990, 2000 and 2010 was shown in this study, and the land use in 1975 was thought to be the substitute during the reference period (see Fig. 2). We have presented the spatial data including the catchment characteristics (see Figs. 1-3) and the precipitation/LUCC-sediment relationships (see Figs. 4-5) as maps.*

10. Comment:

Line 254. As mentioned above, need to be clear about the proxy used for vegetation cover.

Reply: *The LAI was used as the proxy for vegetation cover (see P.12, Lines 357-365).*

11. Comment:

Line 260, 263. Average annual LAI?

Reply: *The growing season (April-October) LAI was used as nearly all the sediment yielded during this period (see P.12, Lines 357-365).*

12. Comment:

Line 278-288. Why use the coefficient of variation and not standard deviation? What do these results tell us?

Reply: *The coefficient of variation is equal to the ratio of standard deviation and average value, and it is better to compare the inter-annual variability of precipitation, streamflow and sediment load among the catchments with distinct different average value (see P.13, Lines 410-412). The results indicated that both the annual value and variability of precipitation decreased, and the annual value of streamflow and sediment load decreased significantly, whereas their inter-annual variability presented somewhat increase, especially for sediment load. The above results indicate the substantially different behaviors of the changes among precipitation, streamflow and sediment load (see P.14, Lines 429-430).*

13. Comment:

Section 3.3. In Equation 6, precipitation is also a temporally variable component, and 'area treated with erosion control measures' is also a spatially variable component. And it seems as though other factors (steeper slopes, soil properties, impermeable surface area, etc) may also play a role in affecting SSY. Moreover, it seems likely that changes in precipitation and land-use change may interact to affect sediment yield. The authors may want to rethink the way they've framed their analysis. Especially as 6/14 catchments in their analysis exhibited no significant correlation. A multiple regression analysis with an interaction term may be a more appropriate means of analysis.

Reply: *First, we have used new regression model to analyze the precipitation-sediment relationship and it improved much compared to the original linear regression model (see response to the main comment #1 of reviewer #1).*

*Second, we have deleted Eq. (6) and reframed the analysis about the spatio-temporal patterns of the impacts of precipitation and LUCC on sediment yield (see the fourth point of the response to the main comment #1 of reviewer #1). Both the temporally variability of precipitation and spatial variability of*

*fraction of area treated with erosion control measures were included in the framework. Furthermore, we have also investigated the possible effects of catch characteristics on the changes of sediment yield (see the second point of the response to #2 main comment of reviewer #1).*

14. Comment:

Lines 387 and throughout: Authors often discuss a 'clear spatial pattern' present in their results, thus maps would be more useful as figures than scatter plots.

Reply: *We have used maps as figures (see Figs. 1-5).*

15. Comment:

Line 393 and 419. Remove undoubtedly.

Reply: *We have removed it (see P.18, Line 716).*

16. Comment:

Lines 449-454. Not quite sure how this resulting empirical relationship follows from the preceding analysis. What are k0 and k1. Also, once better explained, the authors could prove this empirical relationship is robust by showing how accurately it predicts SSY when they input observational data.

Reply: *We have deleted this empirical equation and the related statements.*

17. Comment:

Table 2: Add an ID column.

Reply: *The ID column has been added in Table 2.*

If you have any further questions about this revision, please contact us.

Sincerely Yours,

Dr. Guangyao Gao (gygao@rcees.ac.cn)

Pro. Bojie Fu (bjf@rcees.ac.cn)

Pro. Murugesu Sivapalan (sivapala@illinois.edu)

[revised manuscript text omitted]

---

## Referee Report (RR1)

**Paper title**: Spatio-temporal patterns of the effects of precipitation variability and land use/cover changes on long-term changes in sediment yield in the Loess Plateau, China

**Authors**: Guangyao Gao, Jianjun Zhang, Yu Liu, Zheng Ning, Bojie Fu, and Murugesu Sivapalan

**General comments**

The papers is based on a desktop study of the "Spatio-temporal patterns of the effects of precipitation variability and land use/cover changes on long-term changes in sediment yield in the Loess Plateau, China" i.e. no measurements were performed by the authors. This is, in my own opinion, a very exciting study as it attempted to decouple impacts of two very important controls of sediment generation at catchment scale. Several data sources were consulted and a number of analyses performed, in my own opinion, very well. Great detail is provided and was easy to follow what the authors did to the data collected. This paper will add useful information to the body of knowledge on one of the most important river basins in the world insofar as sedimentation is concerned and should be supported to get it published. Apart from the too many errors, mostly of a grammar nature (understandably most of the authors appear to be non-native English speakers), the authors can do with summaries to some parts of the results presentations. In addition, the structure of the paper, especially the omission of a separate Discussion section, cast further doubts on whether due internal editing of the draft was done before submission for peer review. If the idea was to combine result presentation and discussion, then some parts lack adequate discussion of the results.

Overall, I see this as an important paper which should be published after corrections and improvements as will be indicated below.

**Specific comments**

**Abstract**

Line 25: Insert the exact study period in that sentence, i.e. 1961-2011, not just 50 years.

**Introduction**

Line 75: … in China. This is the …: combine these two sentences as "… in China, which is the …"

Line 80: put comma (,) after SWCM

Line 81: put comma (,) after "reestablishment"

Line 83: … on slopes exceeding …; replace "implemented" by "launched or started"

Line 85: no double full stop, "i.e. ..." should be "i.e."

Line 87-88: … hydrological regimes of the LP in combination with …

Line 89: … declining trend …

Line 93: … contribution of …

Line 94: … between 64 and 89% …

Line 95: what kind of results did Zhao et al (2017) get? Just present a summary like you did for the other references used in this paragraph.

Line 96: Zhang et al (2016) pointed that …

Line 100: … between the 1970s and 1990s …

Line 102: … of these studies …

Line 105: Sun Q et al., 2015; Sun W et al., 2015, please make one "a" and the other one "b" and then remove their initials i.e. Sun et al., 2015a; 2015b, even if they are different people. I think this is better than present their initials.

Line 108: They will also …

Line 112: … region (Figure 1) located in the … LP. The CSHC supplied …

Line 113-4: This region was the focus of our …

Line 115: … of this study were, therefore, to …

Line 118-9: Move "from 15 catchments within the region" to Line 114 between "… sediment load" and "within …" to read "… sediment load from 15 catchments within the region within …"

Line 124: … the Toudaoguai and Longmen …

Line 126: … long and its drainage catchment covers $12.97 \times 10^4$ km$^2$, which is …

Line 128: … precipitation in the region during 1961-2011 was …

Line 129: rearrange to read "… varied from lower than 300 in the northwest to 580 mm in the …"

Line 133: delete CSHC to read "… within the region …"

Line 136: … along a north-south …

Line 139: put a reference … hydrological data during 1961-2011 (REFERENCE). Characteristics …

Line 140: … catchments are presented in Table 1 and Figure 2, showing that the catchments …

Line 142: … (#1-6) had relatively …

Line 145: … (#7-14) were …

Line 148: delete CSHC

Line 151: delete CSHC

Line 152: delete CSHC

Line 156: **2.2 Data collection**

Line 159: delete CSHC, and insert (Figure 1) i.e.  … region (Figure 1) were obtained …

Line 162: The hydro-meteorological data …

Line 167: is it catchment slope gradient?

Line 177: delete **2.2 Methods**
Line 178: **2.3 Trend test**

Line 195: … this method, a regression …

Line 212: This was, therefore, used …

Line 223: Figure 4 shows …

Line 224: delete CSHC

Line 225: … whole area was occupied by …

Line 226-7: … was no significant change …

Line 232: … (Figure 4). In the period 1975 to 2000, …

Line 239-40: … terraces were seen … control projects were …

Line 241: Although the area utilized for engineering …

Line 242: … they immediately and substantially …

Line 244: … catchment area) increased from …

Line 246: … watersheds and the 2000s …

Line 247-49: Some decreases … destroyed". Please do not keep the readers curious here, may you give more light to what happened.

Line 250: delete CSHC

Line 255-7: In the period from 1982-1990 to … increase of LAI … sub-catchments was …

Line 260: … during the period 1961-2011.

Line 261-2: Jialu did not decrease, isn't it?

Line 267-8: The corresponding Q, SSY, SC and C for the whole region were …

Line 270: … whole region were …

Line 273: … and Cs for the fifteen …

Line 276: … with distinctly different …

Line 290: … indicate substantially different …

Line 297: On average, LUCC …

Line 298-300: … period-2, with their respective contributions to sediment load reduction from the reference period to period-3 being 88.67 and 11.33%.

Line 301: … than in period-2 as …

Line 305: … increased and thus the contribution …

Line 314: Correlations between the potential factors …

Line 317: … (see Table 4) showed that check-dam …

Line 318: … period to period-2. Pasture ….

Line 319: … acted as the dominant …

Line 322-324: Based on the above results, the variation … depended on precipitation in the reference period before LUCC took effect and any spatial … of SSY in the catchments were controlled by …

Line 325-7: … (period-2 and period-3) when increased LUCC had taken effect … considerably. The decrease of … insignificant and LUCC contributed over …

Line 328-9: … SSY depended more on … surface area and … possibly played a secondary …

Line 330: … yield was dependent …

Line 331-2: … framework, data were next analysed to generate … patterns constituting respective …

Line 335: … Table 3. The spatial …

Line 337: … period, most of the … How many? I think you can state the figure here.

Line 339: … was significant in eleven … Is it? I see 10!

Lines 340-342: Move "Overall, the regressed … yield changes" to line 353, immediately after "… group of $0.2<a<0.3$".

Line 352: … the Shiwang … I do not understand this.

Line 355: … as indicated by lower $R^2$ …

Line 356: … values in Table 3. The slopes of the regression lines in the …

Line 357-8: … except in Huangfu, Gushan and Kuye which increased slightly.

Line 360: no double full stops … (Figure 9a and 9b).

Line 362-3: … yield were weaker compared to the reference …

Line 364: … relationships between … yield were not significant in all the …

Line 365-7: The slopes of the regression lines during period-3 decreased sharply (Table 3). Six catchments (five in the …part) had negative regression slopes (Figure 9c).

Line 368: … precipitation decreased greatly …

Line 369-70: … did not lead to increased sediment …

Line 371: … during period-3 were clearly different from … period and period-2 (compare Figure 9c against Figure 9a-b).

Lines 375-458: Sounded like a mixture of result presentation and discussion. You need to decide on a style of writing that is consistent throughout the paper.

Line 377: … with time. The impacts were …

Line 390: … In order to fully explore …

Line 392: … scenarios would be needed (Ma et al., 2014; …

Line 407: … more detail and taking …

Line 412: … on 25$^{th}$ August …

Line 413: … 10$^{th}$ August …

Line 420: … be considered when investigating …

Line 422: **3.5 Spatio-temporal pattern** … This is the most important from the paper as deduced from the title of this draft paper; therefore, readers expect it to come early in the discussion.

Line 423: Why did the results drift from CSHC to LP? This study is about CSHC, not LP; even though CSHC is located within LP. I see some kind of mix-up starting with this section.

Line 430: … area to retain precipitation … What do you mean? I think there is a better way of saying what you are trying to say.

Line 437: … in the LP, which trapped … Is it about LP now? Are the results presented here on LP or CSHC?

Line 439-40: … progressively filled with … restoration played a greater role in controlling soil erosion.

Line 441: In order to quantify the effects …

Line 443: … 15 catchments were analysed …

Line 447: The correlations were …

Line 449: … correlations between sediment coefficients and conservation measures were stronger …

Line 456: … respectively. Half …

Lines 460-464: "The Loess … insignificant" is just a summary of what happened in the LP and not a conclusion on the study results. I suggest suppressing this or moving elsewhere.

Line 465: … study has shown that long-term ….

Line 466: … located in the CSHC region.

Line 468: … and landscape controls for the period 1961-2011.

Line 469: suppress "(1961-2011)"

Line 472-5: … measures, there were major reductions in streamflow (65%), sediment yield (88%), sediment concentration (68%) and sediment efficiency, i.e. annual sediment yield/annual precipitation (86%) over the entire 50-year period.

Line 476: … catchments also exhibited interesting …

Line 478: Before LUCC took effect, the data indicates …

Line 485-6: … yield, thus … controls.

Line 487: … linear decreasing …

---

## Author Response (AR2)

Memorandum

To:        Pro. Nunzio Romano, Editor of *Hydrology and Earth System Sciences*

Subject:   **Revision of hess-2016-654**

Dear Pro. Nunzio Romano:

Upon your request, we have carefully addressed all the comments made by the anonymous reviewer on our manuscript (hess-2016-654) entitled "Spatio-temporal patterns of the effects of precipitation variability and land use/cover changes on long-term changes in sediment yield in the Loess Plateau, China" and revised the manuscript accordingly. The detailed comments have helped us further improve the overall quality of the manuscript. The following is the point-point response to all the comments. The page and line numbers in the following response refer to the revised manuscript with changes marked.

Response to Anonymous Referee #1:

General comments:

The papers is based on a desktop study of the "Spatio-temporal patterns of the effects of precipitation variability and land use/cover changes on long-term changes in sediment yield in the Loess Plateau, China" i.e. no measurements were performed by the authors. This is, in my own opinion, a very exciting study as it attempted to decouple impacts of two very important controls of sediment generation at catchment scale. Several data sources were consulted and a number of analyses performed, in my own opinion, very well. Great detail is provided and was easy to follow what the authors did to the data collected. This paper will add useful information to the body of knowledge on one of the most important river basins in the world insofar as sedimentation is concerned and should be supported to get it published. Apart from the too many errors, mostly of a grammar nature (understandably most of the authors appear to be non-native English speakers), the authors can do with summaries to some parts of the results presentations. In addition, the structure of the paper, especially the omission of a separate Discussion section, cast further doubts on whether due internal editing of the draft was done before submission for peer review. If the idea was to combine result presentation and discussion, then some parts lack adequate discussion of the results.

Overall, I see this as an important paper which should be published after corrections and improvements as will be indicated below.

Reply: *Thanks very much for the nice comments.*

    *First, we have double checked the writing of the manuscript and addressed all the specific comments about the presentation of the paper (see the following point-to-point replies to the comments). The revision further polished the manuscript.*

    *Second, we have changed the structure of the manuscript and separated the results and discussion. The statements about the reasons for the spatial-temporal patterns of the impacts of precipitation variability (especially the role of within-year rainfall patterns and storm events) and land*

*use/cover changes on sediment yield variations were moved into the "3.6 Discussion" section (see P.22, Line 478 to P.25, Line 533).*

*It is necessary to point out that the first author finished this work through collaboration with Pro. Murugesu Sivapalan who hosted the visit of the first author in University of Illinois at Urbana-Champaign during June 2016-June 2017. Pro. Murugesu Sivapalan edited the original and revised versions of the manuscript for three times to guarantee the quality of writing.*

Specific comments:

**Abstract**
Line 25: Insert the exact study period in that sentence, i.e. 1961-2011, not just 50 years.

Reply: *The revegetation efforts and engineering measures have been implemented since the 1950s in the Loess Plateau (see), and the study period of this study was 1961-2011 (see P.1, Line 25).*

**Introduction**
Line 75: … in China. This is the …: combine these two sentences as "… in China, which is the …"

Reply: *Done (see P.4, Line 75).*

Line 80: put comma (,) after SWCM

Reply: *Done (see P.4, Line 81).*

Line 81: put comma (,) after "reestablishment"

Reply: *Done (see P.4, Line 82).*

Line 83: … on slopes exceeding …; replace "implemented" by "launched or started"

Reply: *Done (see P.4, Line 85).*

Line 85: no double full stop, "i.e. ..." should be "i.e."

Reply: *Done (see P.4, Line 86).*

Line 87-88: … hydrological regimes of the LP in combination with …

Reply: *Done (see P.4, Lines 88-89).*

Line 89: … declining trend …

Reply: *Done (see P.4, Line 90).*

Line 93: … contribution of …

Reply: *Done (see P.5, Line 94).*

Line 94: … between 64 and 89% …

Reply: *Done (see P.5, Line 95).*

Line 95: what kind of results did Zhao et al (2017) get? Just present a summary like you did for the other references used in this paragraph.

Reply: *We have given the summary results of Zhao et al. (2017) (see P.5, Lines 97-100).*

Line 96: Zhang et al (2016) pointed that …

Reply: *Done (see P.5, Line 100).*

Line 100: … between the 1970s and 1990s …

Reply: *Done (see P.5, Line 104).*

Line 102: … of these studies …

Reply: *Done (see P.5, Line 107).*

Line 105: Sun Q et al., 2015; Sun W et al., 2015, please make one "a" and the other one "b" and then remove their initials i.e. Sun et al., 2015a; 2015b, even if they are different people. I think this is better than present their initials.

Reply: *Done (see P.5, Lines 110-111).*

Line 108: They will also …

Reply: *Done (see P.6, Line 114).*

Line 112: … region (Figure 1) located in the … LP. The CSHC supplied …

Reply: *Done (see P.6, Line 118).*

Line 113-4: This region was the focus of our …

Reply: *Done (see P.6, Line 120).*

Line 115: … of this study were, therefore, to …

Reply: *Done (see P.6, Line 122).*

Line 118-9: Move "from 15 catchments within the region" to Line 114 between "… sediment load" and "within …" to read "… sediment load from 15 catchments within the region within …"

Reply: *Done (see P.6, Lines 121 and 125).*

Line 124: … the Toudaoguai and Longmen …
Reply: *Done (see P.6, Line 131).*

Line 126: … long and its drainage catchment covers 12.97x10$^4$ km$^2$, which is …
Reply: *Done (see P.6, Lines 133-134).*

Line 128: … precipitation in the region during 1961-2011 was …
Reply: *Done (see P.6, Line 135 to P.7, Line 136).*

Line 129: rearrange to read "… varied from lower than 300 in the northwest to 580 mm in the …
Reply: *Done (see P.7, Lines 136-137).*

Line 133: delete CSHC to read "… within the region …"
Reply: *Done (see P.7, Line 141).*

Line 136: … along a north-south …
Reply: *Done (see P.7, Line 144).*

Line 139: put a reference … hydrological data during 1961-2011 (REFERENCE). Characteristics …
Reply: *References have been put (see P.7, Lines 147-148).*

Line 140: … catchments are presented in Table 1 and Figure 2, showing that the catchments …
Reply: *Done (see P.7, Lines 148-149).*

Line 142: … (#1-6) had relatively …
Reply: *Done (see P.7, Line 150).*

Line 145: … (#7-14) were …
Reply: *Done (see P.7, Line 154).*

Line 148: delete CSHC
Reply: *Done (see P.7, Line 156).*

Line 151: delete CSHC

Reply: *Done (see P.8, Line 159).*

Line 152: delete CSHC

Reply: *Done (see P.8, Line 160).*

Line 156: **2.2 Data collection**

Reply: *Done (see P.8, Line 164).*

Line 159: delete CSHC, and insert (Figure 1) i.e. … region (Figure 1) were obtained …

Reply: *Done (see P.8, Line 167).*

Line 162: The hydro-meteorological data …

Reply: *Done (see P.8, Line 170).*

Line 167: is it catchment slope gradient?

Reply: *Yes (see P.8, Line 175).*

Line 177: delete **2.2 Methods**

Reply: *Done (see P.9, Line 185).*

Line 178: **2.3 Trend test**

Reply: *Done (see P.9, Line 186).*

Line 195: … this method, a regression …

Reply: *Done (see P.10, Line 203).*

Line 212: This was, therefore, used …

Reply: *Done (see P.10, Line 220).*

Line 223: Figure 4 shows …

Reply: *Done (see P.11, Line 231).*

Line 224: delete CSHC

Reply: *Done (see P.11, Line 232).*

Line 225: … whole area was occupied by …

Reply: *Done (see P.11, Line 233).*

Line 226-7: … was no significant change …

Reply: *Done (see P.11, Line 235).*

Line 232: … (Figure 4). In the period 1975 to 2000, …

Reply: *Done (see P.11, Line 240).*

Line 239-40: … terraces were seen … control projects were …

Reply: *Done (see P.12, Lines 247-248).*

Line 241: Although the area utilized for engineering …

Reply: *Done (see P.12, Line 249).*

Line 242: … they immediately and substantially …

Reply: *Done (see P.12, Line 250).*

Line 244: … catchment area) increased from …

Reply: *Done (see P.12, Line 252).*

Line 246: … watersheds and the 2000s …

Reply: *Done (see P.12, Line 255).*

Line 247-49: Some decreases … destroyed". Please do not keep the readers curious here, may you give more light to what happened.

Reply: *We have given more statements to make it clear (see P.12, Lines 255-259).*

Line 250: delete CSHC

Reply: *Done (see P.12, Line 260).*

Line 255-7: In the period from 1982-1990 to … increase of LAI … sub-catchments was …

Reply: *Done (see P.12, Lines 265-267).*

Line 260: … during the period 1961-2011.

Reply: *Done (see P.13, Lines 270-271).*

Line 261-2: Jialu did not decrease, isn't it?

Reply: *Yes, the annual precipitation in the Jialu catchment increased (see P.13, Lines 271-272).*

Line 267-8: The corresponding Q, SSY, SC and C for the whole region were …

Reply: *Done (see P.13, Lines 278-279).*

Line 270: … whole region were …

Reply: *Done (see P.13, Lines 280-281).*

Line 273: … and Cs for the fifteen …

Reply: *Done (see P.13, Line 284).*

Line 276: … with distinctly different …

Reply: *Done (see P.13, Line 287).*

Line 290: … indicate substantially different …

Reply: *Done (see P.14, Line 301).*

Line 297: On average, LUCC …

Reply: *Done (see P.14, Line 309).*

Line 298-300: … period-2, with their respective contributions to sediment load reduction from the reference period to period-3 being 88.67 and 11.33%.

Reply: *Done (see P.14, Line 310 to P.15, Line 312).*

Line 301: … than in period-2 as …

Reply: *Done (see P.15, Line 313).*

Line 305: … increased and thus the contribution …

Reply: *Done (see P.15, Line 317).*

Line 314: Correlations between the potential factors …

Reply: *Done (see P.15, Line 326).*

Line 317: … (see Table 4) showed that check-dam …

Reply: *Done (see P.15, Line 329).*

Line 318: … period to period-2. Pasture ….

Reply: *Done (see P.15, Line 330).*

Line 319: … acted as the dominant …

Reply: *Done (see P.15, Line 331).*

Line 322-324: Based on the above results, the variation … depended on precipitation in the reference period before LUCC took effect and any spatial … of SSY in the catchments were controlled by …

Reply: *Done (see P.16, Lines 334-336).*

Line 325-7: … (period-2 and period-3) when increased LUCC had taken effect … considerably. The decrease of … insignificant and LUCC contributed over …

Reply: *Done (see P.16, Lines 338-340).*

Line 328-9: … SSY depended more on … surface area and … possibly played a secondary …

Reply: *Done (see P.16, Lines 341-342).*

Line 330: … yield was dependent …

Reply: *Done (see P.16, Line 343).*

Line 331-2: … framework, data were next analysed to generate … patterns constituting respective …

Reply: *Done (see P.16, Line 344-345).*

Line 335: … Table 3. The spatial …

Reply: *Done (see P.16, Line 348).*

Line 337: … period, most of the … How many? I think you can state the figure here.

Reply: *There were eleven catchments with significant correlation between precipitation and sediment yield (see P.16, Lines 350-353).*

Line 339: … was significant in eleven … Is it? I see 10!

Reply: *The correlation between precipitation and sediment yield was significant in eleven catchments (catchments #1-#7, #10, #12, #13 and #15, see Table 3).*

Lines 340-342: Move "Overall, the regressed … yield changes" to line 353, immediately after "… group of 0.2<a<0.3".

Reply: *Done (see P.16, Lines 353-355, P.17, Lines 367-370).*

Line 352: … the Shiwang … I do not understand this.

Reply: *The slope of regression equation between annual precipitation and sediment yield was lowest in the Shiwang catchment compared to other fourteen catchments (see P.17, Lines 365-366).*

Line 355: … as indicated by lower $R^2$ …

Reply: *Done (see P.17, Line 372).*

Line 356: … values in Table 3. The slopes of the regression lines in the …

Reply: *Done (see P.17, Line 373).*

Line 357-8: … except in Huangfu, Gushan and Kuye which increased slightly.

Reply: *Done (see P.17, Lines 374-375).*

Line 360: no double full stops … (Figure 9a and 9b).

Reply: *Done (see P.17, Line 377).*

Line 362-3: … yield were weaker compared to the reference …

Reply: *Done (see P.18, Lines 380-381).*

Line 364: … relationships between … yield were not significant in all the …

Reply: *Done (see P.18, Line 382).*

Line 365-7: The slopes of the regression lines during period-3 decreased sharply (Table 3). Six catchments (five in the …part) had negative regression slopes (Figure 9c).

Reply: *Done (see P.18, Lines 383-385).*

Line 368: … precipitation decreased greatly …

Reply: *Done (see P.18, Line 386).*

Line 369-70: … did not lead to increased sediment …

Reply: *Done (see P.18, Line 388).*

Line 371: … during period-3 were clearly different from … period and period-2 (compare Figure 9c against Figure 9a-b).

Reply: *Done (see P.18, Lines 389-391).*

Lines 375-458: Sounded like a mixture of result presentation and discussion. You need to decide on a style of writing that is consistent throughout the paper.

Reply: *We have changed the structure of the manuscript and separated the results and discussion. The statements about the reasons for the spatial-temporal patterns of the impacts of precipitation variability (especially the role of within-year rainfall patterns and storm events) and land use/cover changes on sediment yield variations were moved into the "3.6 Discussion" section (see P.22, Line 478 to P.25, Line 533).*

Line 377: … with time. The impacts were …

Reply: *Done (see P.18, Line 396).*

Line 390: … In order to fully explore …

Reply: *Done (see P.22, Line 483).*

Line 392: … scenarios would be needed (Ma et al., 2014; …

Reply: *Done (see P.22, Line 485).*

Line 407: … more detail and taking …

Reply: *Done (see P.23, Line 500).*

Line 412: … on 25th August …

Reply: *Done (see P.23, Line 505).*

Line 413: … 10th August …

Reply: *Done (see P.23, Line 506).*

Line 420: … be considered when investigating …

Reply: *Done (see P.24, Line 513).*

Line 422: **3.5 Spatio-temporal pattern** … This is the most important from the paper as deduced from the title of this draft paper; therefore, readers expect it to come early in the discussion.

Reply: *We have changed the structure of the manuscript and separated the results and discussion. Through this revision, section 3.5 moved forward (see P.20, Line 441 to P.22, Line 477).*

Line 423: Why did the results drift from CSHC to LP? This study is about CSHC, not LP; even though CSHC is located within LP. I see some kind of mix-up starting with this section.

Reply: *We have focused on the CSHC in this section, and presented the results in only CSHC (see P.24, Line 515 to P.25, Line 533).*

Line 430: … area to retain precipitation … What do you mean? I think there is a better way of saying what you are trying to say.

Reply: *We have revised this sentence to make it clear (see P.24, Lines 522-523).*

Line 437: … in the LP, which trapped … Is it about LP now? Are the results presented here on LP or CSHC?

Reply: *We have presented the results about the sediment trapping of check-dams in the CSHC (see P.24, Lines 529-531).*

Line 439-40: … progressively filled with … restoration played a greater role in controlling soil erosion.

Reply: *Done (see P.25, Lines 532-533).*

Line 441: In order to quantify the effects …

Reply: *Done (see P.21, Line 460).*

Line 443: … 15 catchments were analysed …

Reply: *Done (see P.21, Line 462).*

Line 447: The correlations were …

Reply: *Done (see P.22, Line 466).*

Line 449: … correlations between sediment coefficients and conservation measures were stronger …

Reply: *Done (see P.22, Line 468).*

Line 456: … respectively. Half …

Reply: *Done (see P.22, Line 475).*

Lines 460-464: "The Loess … insignificant" is just a summary of what happened in the LP and not a conclusion on the study results. I suggest suppressing this or moving elsewhere.

Reply: *We have deleted these sentences (see P.25, Lines 535-539).*

Line 465: … study has shown that long-term ….

Reply: *Done (see P.25, Line 540).*

Line 466: … located in the CSHC region.

Reply: *Done (see P.25, Line 542).*

Line 468: … and landscape controls for the period 1961-2011.

Reply: *The study period (1961-2011) has been given in P.25, Line 542.*

Line 469: suppress "(1961-2011)"

Reply: *Done (see P.25, Line 546).*

Line 472-5: … measures, there were major reductions in streamflow (65%), sediment yield (88%), sediment concentration (68%) and sediment efficiency, i.e. annual sediment yield/annual precipitation (86%) over the entire 50-year period.

Reply: *Done (see P.25, Lines 549-552).*

Line 476: … catchments also exhibited interesting …

Reply: *Done (see P.25, Line 553).*

Line 478: Before LUCC took effect, the data indicates …

Reply: *Done (see P.26, Line 556).*

Line 485-6: … yield, thus … controls.

Reply: *Done (see P.26, Lines 562-563).*

Line 487: … linear decreasing …

Reply: *Done (see P.26, Line 565).*

If you have any further questions about this revision, please contact us.

Sincerely Yours,

Dr. Guangyao Gao (gygao@rcees.ac.cn)

Pro. Bojie Fu (bjf@rcees.ac.cn)

Pro. Murugesu Sivapalan (sivapala@illinois.edu)

[revised manuscript text omitted]